



# PAMTRA 1.0: A Passive and Active Microwave radiative TRAnsfer tool for simulating radiometer and radar measurements of the cloudy atmosphere

Mario Mech[a], Maximilian Maahn[b,c], Stefan Kneifel[a], Davide Ori[a], Emiliano Orlandi[d, a],
Pavlos Kollias[e, a], Vera Schemann[a], and Susanne Crewell[a]

[a]Institute for Geophysics and Meteorology, University of Cologne, Cologne, Germany
[b]Cooperative Institute for Research in Environmental Sciences, University of Colorado Boulder, Boulder, CO, USA
[c]Physical Sciences Division, NOAA Earth System Research Laboratory, Boulder, CO, USA
[d]Radiometer-Physics GmbH, Meckenheim, Germany
[e]School of Marine and Atmospheric Sciences, Stony Brook University, NY, USA

**Correspondence:** Mario Mech (mario.mech@uni-koeln.de)

**Abstract.** Forward models are a key tool to generate synthetic observations given the knowledge of the atmospheric state. In this way they are an integral part of inversion algorithms that aim to retrieve geophysical variables from observations or in data assimilation. Their application for the exploitation of the full information content of remote sensing observations becomes increasingly important when these are used to evaluate the performance of cloud resolving models (CRMs). Herein,
CRMs profiles or fields provide the input to the forward model whose simulation results are subsequently compared to the observations. This paper introduces the freely available comprehensive microwave forward model PAMTRA (Passive and Active Microwave TRAnsfer), demonstrates its capabilities to simulate passive and active measurements across the microwave spectral region for up- and downward looking geometries, and illustrates how the forward simulations can be used to evaluate CRMs and to interpret measurements to improve our understanding of cloud processes.

PAMTRA is unique as it treats passive and active radiative transfer (RT) in a consistent way with the passive forward model providing up- and down-welling polarized brightness temperatures and radiances for arbitrary observation angles. The active part is capable of simulating the full radar Doppler spectrum and its moments. PAMTRA is designed to be flexible with respect to instrument specifications, interfaces to many different formats of in- and output type, especially CRMs, spanning the range from bin-resolved microphysical output to one- and two-moment schemes, and to in situ measured hydrometeor properties. A
specific highlight is the incorporation of the self-similar Rayleigh–Gans Approximation (SSRGA) both for active and passive applications which becomes especially important for the investigation of frozen hydrometeors.



# 1 Introduction

The use of passive and active microwave sensors in atmospheric research has experienced rapid growth in the last decades due to their unique ability to provide information on clouds and precipitation as well as for thermodynamic profiling even under cloudy conditions. Passive and active microwave sensors are highly complementary and are therefore often collocated on space-, airborne, or ground-based observing platforms. The strength of this combination is based on the ability of the radar to provide very detailed information about the vertical structure of hydrometeors, in-cloud dynamics (Borque et al., 2016), as well as microphysical processes (Kalesse et al., 2016), while the passive sensors add information on thermodynamic profiles, constrain column integrated hydrometeor quantities, and provide wide swath information from satellite. Prominent examples for combined satellite sensors are the Tropical Rainfall Measurement Mission (TRMM; Kummerow et al., 1998), the Global Precipitation Mission (GPM; Hou et al., 2014), and the Afternoon-Train (A-Train; L'Ecuyer and Jiang, 2011). Passive and active microwave instruments are also commonly combined in airborne observatories such as the High Altitude and LOng range research aircraft (HALO) Microwave Package (HAMP; Mech et al., 2014) or the remote sensing package of the Wyoming King Air (Wang et al., 2012). From ground, the standard configuration needed to determine detailed cloud vertical profile information (e.g., the CloudNet algorithm (Illingworth et al., 2015)) at several ground-based super-sites (Mather and Voyles, 2013; Löhnert et al., 2015) includes a cloud radar, a microwave radiometer, and a ceilometer.

To fully exploit remote sensing measurements, radiative transfer (RT) models are needed which convert an atmospheric state into a synthetic measurement. They are key tools for the design of new sensors, for the development of retrieval algorithms and the improvement of atmospheric models both for numerical weather prediction (NWP) and climate applications. Herein, a particular challenge is the realistic description of hydrometeors' and their particle size distributions (PSD) as well as their respective single scattering properties (Petty, 2001), which are required when solving the RT equation. Specifically, an accurate but also computationally efficient description of the scattering properties of ice and snow particles for global applications is needed (Geer and Baordo, 2014). Closure studies in which detailed in-situ measurements of hydrometeor properties and corresponding remote sensing measurements are connected with the help of an RT model can improve our knowledge of these interaction processes (e.g., Tridon et al., 2019).

The connection between atmospheric models and RT is twofold. On the one hand realistic representations of the atmospheric state with emphasis on hydrometeors are needed as input for the RT together with instrument models to yield synthetic measurements (e.g., sampling geometry, noise characteristics). Thus, cloud resolving models (CRMs) are frequently used as RT input in design studies and retrieval development (Chaboureau et al., 2008; Matsui et al., 2013). On the other hand, the remote sensing measurements shall be used for improving the atmospheric models. Most directly, measurements are used in data assimilation together with fast RT operators for NWP models. This is especially demanding under cloudy conditions but of growing importance for NWP (Geer et al., 2017). To improve the representation of clouds and precipitation in atmospheric models in general, microphysical schemes are under development with a tendency towards increasing the number of hydrometeor categories and PSD moments incorporated. Microwave measurements are well suited for evaluating the performance of





these schemes but a thorough matching of the predicted hydrometeor properties and the assumptions in the RT needs to be realized (e.g., Han et al., 2013; Matsui et al., 2013; Schemann and Ebell, 2019)

In summary, various applications require RT models to be flexible in respect to the adaptation of the given hydrometeor information and the different instrument specifics. Here we present the Passive and Active Microwave radiative TRAnsfer

operator (PAMTRA) which has been developed exactly for this purpose. Along with the increasing use of microwave remote sensing several RT models have already been developed in the past. In the following we give examples of important RT models to provide context to our motivation to develop a new RT framework.

The Radiative Transfer for the TIROS Operational Vertical Sounder (TOVS) (RTTOV; Saunders et al., 1999, 2018) has been developed for the specific application of NWP data assimilation to respond to the requirement of high computational

performance. For this purpose, RTTOV employs parameterizations tailored to specific microwave satellite radiometers. It provides the tangent linear, adjoint and Jacobian matrix to enable all-sky data assimilation. Recently, RTTOV-gb has been released which also allows to simulate ground-based sensors (De Angelis et al., 2016). Similar to RTTOV the Community Radiative Transfer Model (CRTM; Ding et al., 2011) has been developed to efficiently simulate specific sensors (e.g., satellite instrumentation).

For developing the parameterizations for the fast RT models, i.e., determining sensor specific coefficients, reference RT simulations with line-by-line models are needed. These are typically one-dimensional models that assume a plane-parallel atmosphere and need information on gaseous absorption and hydrometeor single scattering properties for each vertical layer. For example, AMSUTRAN (Turner et al., 2019) calculates profiles of layer-to-space transmittances as the basis for the training of RTTOV. It includes absorption routines based on the Millimeter-wave Propagation Model (MPM; Liebe et al., 1991, 1993)

with subsequent spectroscopic modifications.

The RT can be principally solved if gas absorption and single scattering properties for hydrometeors are specified. Especially, a realistic representation of single-scattering properties of frozen particles is still a challenge for any RT. The number of databases including scattering properties of various habits, densities, orientations, and composition is rapidly increasing during recent years (Kneifel et al., 2018). However, many RT are still using spheroidal approximations due to their low computational

costs and flexibility to account for particle properties such as mass–size relation. New approximations, which take the fractal properties of aggregates better into account became recently available (Hogan and Westbrook, 2014; Hogan et al., 2017).

Two widely used codes for polarized microwave radiation are the RT3 and RT4 models provided by Evans and Stephens (1991, 1995, 2010). RT3 solves RT for atmospheres with randomly oriented particles; RT4 is an extension of RT3 and also accepts azimuthal symmetric oriented particles. Several RT models implemented these codes as the RT solving algorithm

(Deiveegan et al., 2008; Buehler et al., 2018) with different options for information on gaseous absorption, or single scattering properties. RT4 is also used for the passive component of the RT framework presented in this manuscript.

For active microwave sensors QuickBeam (Haynes et al., 2007) is able to simulate radar reflectivity profiles for bottom-up and top-down perspective and is part of the CFMIP (Cloud Feedback Model Intercomparison Project) Observation Simulator (COSP; Bodas-Salcedo et al., 2011). Higher Doppler spectral moments (Kollias et al., 2007) and radar polarimetry which are

often provided by ground-based Doppler cloud radars can be calculated with the Cloud Resolving Model Radar Simulator



(CR-SIM; Oue et al., 2019). The Polarimetric radar simulator (POLARRIS) recently presented by (Matsui et al., 2019) is a forward and inverse model for polarimetric radar observables.

Only very few RT provide simultaneous passive/active simulations. Two examples are the Passive and Active Microwave-Vector Radiative Transfer (PAM-VRT; Yang and Min, 2015) and the Atmospheric Radiative Transfer Simulator (ARTS; Eriksson et al., 2011; Buehler et al., 2018). Both RT are suited to simulate spaceborne and ground-based sensors including more complex (non-spheroidal) single scattering databases for frozen particles. However, both RT models do not provide simulations of the full Doppler spectrum.

Though several studies in the past have already used parts of the Passive and Active Microwave TRAnsfer (PAMTRA) tool, e.g., Acquistapace et al. (2017) for optimized drizzle detection, Cadeddu et al. (2019) for ground-based radiometer retrieval in raining conditions, Maahn and Löhnert (2017) for simulations of in situ aircraft measurements, Heinze et al. (2017) and Schemann and Ebell (2019) for CRM evaluation, it has now been converted into a versatile, freely available tool. Herein the main motivation was the need to have a RT tool which can simulate microwave as well as Doppler radars for ground-based, airborne, or spaceborne platforms using state-of-the-art scattering models. PAMTRA provides passive and active RT simulations in a consistent way for a plan-parallel, one-dimensional, horizontally homogeneous atmosphere with hydrometeors for up- and down-welling microwave radiances. A particular focus in the design of PAMTRA was put on providing maximum flexibility to various model output (one-moment, two-moment, or full-bin schemes) or in situ measured hydrometeor properties. It also was intended to allow the user to select a number of scattering and absorption models for maximum flexibility in the assumptions made in the microphysical parameterizations.

This paper provides a description of the first comprehensive PAMTRA version 1.0 and advocates its use with a range of examples demonstrating its value in investigating cloud and precipitation processes. Section 2 gives an overview of the general architecture of PAMTRA including the description of the passive and active RT. This general part of the RT is followed by descriptions of how atmospheric properties such as gas absorption, particle size distribution, scattering and absorption of hydrometeors, as well as boundary conditions are treated in PAMTRA. It also provides an overview of the wide range of selectable user options, e.g., scattering/absorption models, databases. Application examples (Sect. 3) include ground-based, airborne, and satellite perspectives for passive and active microwave sensors. In Sect. 4 a summary as well future perspectives are given.

## 2   Model framework

PAMTRA is a FORTRAN/Python model framework for the simulation of passive and active RT (including radar Doppler spectra) in a plane-parallel, one-dimensional, and horizontally homogeneous atmosphere for the microwave frequency range. Figure 1 shows a flow diagram of the various steps performed in the FORTRAN core of the present model setup. For the simulation, the model needs various inputs (shown in reddish colors) that describe the atmospheric state, the assumption on absorption, scattering, and surface emissivity, and instrument specifications. Depending on this input, the interaction parameters within various modules (white boxes) are generated. These parameters serve as input for the solving routines for the passive



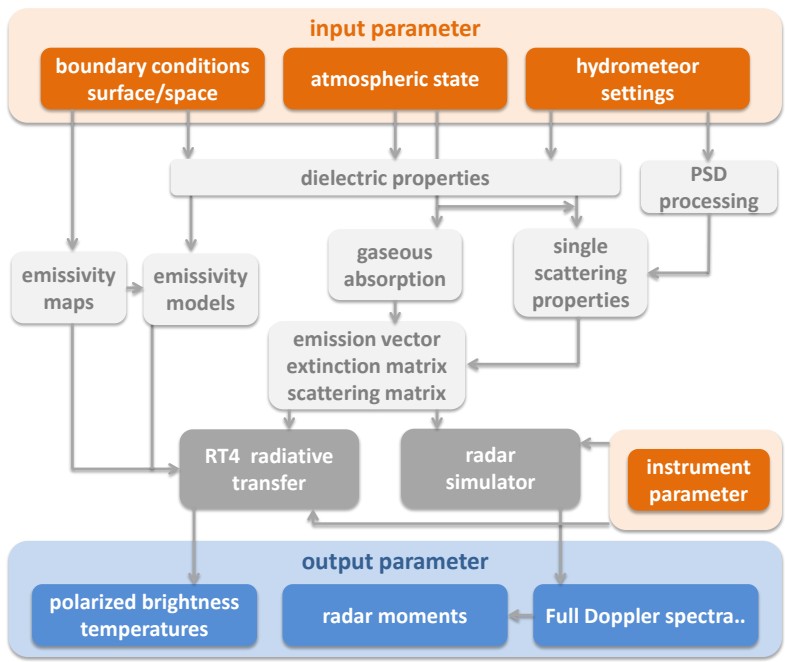

**Figure 1.** Flowchart of the various steps performed during a PAMTRA simulation. Orange areas describe input parameters given by the user with the Python interface or text files. Gray boxes are the FORTRAN model parts where the various interaction parameters are generated and the radiative transfer or the radar simulator get processed. Blue boxes describe the model output.

and active part (shown in gray). The simulations produce polarized radiances or brightness temperatures ($T_B$) for the passive part and radar polarimetric Doppler spectra (and derived moments such as reflectivity, mean Doppler velocity, skewness, and kurtosis, as well as left and right slopes) for the active part. The simulation is performed at any observation geometry (zenith/nadir looking or slanted). Table 1 summarizes the main features of PAMTRA.

5      pyPAMTRA adds a Python framework around the FORTRAN core which allows to call PAMTRA directly from Python without using the FORTRAN I/O routines. Consequently, pyPAMTRA is a more user-friendly way to access the PAMTRA model and is the common way to use the model framework. It includes a collection of supporting routines, e.g., for importing model data or producing file or graphical output of the simulation results. With pyPAMTRA, parallel execution of PAMTRA on multi-core processor machines and clusters is possible. Furthermore, by using Python for I/O and flow control, it is easier 10   to interface PAMTRA to instrument or atmospheric models, as well as post-processing routines.

### 2.1    Microwave radiometer simulator

For the passive part, the monochromatic vector RT equation for an independent column approximation and plane-parallel atmosphere is solved using the RT4 code of Evans and Stephens (1995). The assumption of a plane-parallel geometry and horizontal homogeneity is sufficient for most RT problems in the microwave spectral range with the exception of strongly scat-





tering precipitation schemes where the radiation does not originate within the instruments field of view (Battaglia and Tanelli, 2011). The RT is solved numerically by the doubling and adding method (Liou, 2002, p. 290), which follows the interaction principle. It relates the interaction of radiation with a medium by relating the radiation emerging from an atmospheric layer to the radiation incident upon and generated within this layer. Thereby, homogeneous atmospheric layers are subdivided into

thinner sub-layers such that finite differences can be applied as a good approximation. The transformation from finite differences to the interaction principle is called initialization. By this initialization, the scattering matrices are related to reflection and transmission matrices. The integration over these thin sub-layers is performed by the doubling algorithm. Afterwards, for each output level, the adding algorithm is applied, i.e., transmission, reflection, and emissions of the layers above and below the output layer are added.

As mentioned above, the doubling and adding is done in PAMTRA by RT4. In comparison to the formerly introduced RT3 (Evans and Stephens, 1991), RT4 enables the user to perform polarized RT calculations for non-spherical and oriented particles. Since one of the major goals in developing PAMTRA was to handle hydrometeor interactions as flexible as possible, the possibility to simulate the RT for oriented particles with any shape is mandatory. RT4 calculates polarized $T_B$ (vertical and horizontal) for each discrete quadrature angle and frequency and up- and downward looking geometries at any height within

the atmosphere.

## 2.2   Radar simulator

The PAMTRA radar simulator estimates the full radar Doppler spectrum based on the single scattering properties of each hydrometeor species (see Sect. 2.5.2); it is mainly based on the concepts developed by Oue et al. (2019). First, the backscattering cross section $\sigma_B(D)$ in $\mathrm{m}^2$ of the individual hydrometeor particles with maximum dimension $D$ is converted to the

volumetric back-scattering $\eta_D(D)$ in the unit of spectral radar reflectivity $\mathrm{mm}^6\mathrm{m}^{-3}\mathrm{m}^{-1}$

$$\eta_D(D) = 10^{18}\sigma_B(D)\, n(D)\frac{\lambda^4}{\pi^5|K_w|^2}, \tag{1}$$

where $\lambda$ is the wavelength in m, $n(D)$ is the normalized PSD in $\mathrm{m}^{-4}$, and $|K_w^2|$ is the dielectric factor of water related to the refractive index. It is a common convention to use the value for liquid water at centimeter wavelengths ($|K_w^2| = 0.93$, Ulaby et al., 1981) regardless of whether ice or liquid clouds are observed. Nevertheless, as $|K_w^2|$ also depends on frequency it is

possible to change it for optimal adaptation to a specific problem. However, multiple scattering can affect radar measurements in cases of strong precipitation, short wavelength and large radar footprint (Battaglia et al., 2010). These effects can be particularly relevant for satellite radar observations of strong precipitation at W-band such as those of CloudSat (Matrosov and Battaglia, 2009), but can be neglected for most other cloud radar applications and is not considered by PAMTRA.

   The radar reflectivity factor $Z_e$ can be simply obtained from Eq. 1 by integrating over the PSD. Unlike other radar simulators

(e.g., Haynes et al., 2007; Buehler et al., 2005), PAMTRA provides the option to simulate the full radar Doppler spectrum which is necessary to derive higher radar moments (mean Doppler velocity $MDV$, skewness, kurtosis). Deriving the radar moments from the simulated spectra also allows to account for instrument-specific characteristics such as the minimal sensitivity. Those





instrument characteristics can have an impact on the derived moments and hence are important to take into account when aiming to simulate observations of a specific radar system (see also examples in Sect. 3). The radar Doppler spectra simulator included in PAMTRA is partly based on the concept of Kollias et al. (2011, 2014), and the basic working principle is explained in the following.

As the measured Doppler spectrum is a function of fall velocity rather than particle size, the size descriptor of the spectral radar reflectivity $\eta_D$ is changed from $D$ to fall velocity $v$ with

$$\eta_v(v) = \eta_D(D)\frac{\partial D}{\partial v}. \tag{2}$$

where, after the transformation by the differential $\frac{\partial D}{\partial v}$ (measured in s), the spectral radar reflectivity $\eta_v(v)$ assumes the measuring units of $\mathrm{mm^6 m^{-3} s m^{-1}}$. The user can choose between various hydro-dynamical models to estimate hydrometeor terminal

velocity $v$ and $\frac{\partial D}{\partial v}$ based on their physical properties and environmental conditions such as air density, temperature, and pressure. For liquid drops, PAMTRA uses the relation provided by Khvorostyanov and Curry (2002) as a default. For ice and snow particles, the modified relation by Heymsfield and Westbrook (2010) is recommended because it is in better agreement with recent experiments using ice analogues (Westbrook and Sephton, 2017). For PAMTRA, $v$ is defined such that positive values refer to particles moving towards the radar.

The velocity resolution of $\eta_v(v)$ is related to the bin spacing of the particle size distribution. However, real radar Doppler spectra have boundaries of maximum/minimum Doppler velocity $v_{\mathrm{nyq}} = \mathrm{PRF} * \lambda/4$ determined by the pulse repetition frequency PRF and the radar wavelength $\lambda$. The velocity resolution is determined by the number of Fast Fourier Transform (FFT) points $n_{\mathrm{fft}}$ used to derive the radar Doppler spectrum. These parameters are adjustable in the radar Doppler spectra simulator. $\eta_v(v)$ of all hydrometeors is then linearly interpolated onto the spectral resolution of the simulated radar. Furthermore, if the

fall velocity exceeds $v_{\mathrm{nyq}}$, the simulator adds velocity folding effects (aliasing) to the spectrum.

     In reality, the idealized $\eta_v(v)$ spectrum is affected by dynamical and instrument effects such as attenuation, kinematic broadening, vertical air motion, and radar noise (Doviak and Zrnic, 1993). In PAMTRA, the attenuated $\eta_v(v)$ is obtained by subtracting the cumulative path integrated attenuation, which is estimated from the extinction of gases and hydrometeors depending on measurement geometry (ground-based, airborne, or space-based). Kinematic broadening is assumed to have a

Gaussian distribution that is convoluted with $\eta_v(v)$ to simulate the broadening of the Doppler spectrum due to air motions (Gossard and Strauch, 1989) as described in detail in Maahn et al. (2015). In addition to kinematic broadening, also a constant vertical air motion $V_{\mathrm{air}}$ can be added that shifts the Doppler peak in the velocity spectrum without broadening the peak. To account for the radar receiver noise, the radar receiver noise power $N_P$ (in units of $\mathrm{mm^6 m^{-3}}$ in accordance with $Z_e$) is added to the spectrum $\eta_v(v)$. To account for the loss of radar sensitivity due to range, $N_P$ is scaled with range squared. Because the

noise is assumed to be white, random perturbations are added to every bin $i$ of the spectrum in order to account for random noise effects following (Zrnić, 1975). To make the simulations by the radar simulator reproducible, the random seed used to obtain $r(i)$ can be defined in PAMTRA. Finally, the spectrum is successively averaged $n_{\mathrm{ave}}$ times to account for smoothing and turbulence broadening.





Once the simulation of the non-idealized radar Doppler spectrum is complete, the corresponding moments are estimated similar to a real radar data processing scheme (e.g., Maahn and Kollias, 2012): first, the noise is removed from the spectrum, and second the moments of the hydrometeor peak are determined. In case of several hydrometeor peaks in the same spectrum that are fully separated by the noise floor (multi-modal spectra, e.g., Williams et al., 2018), PAMTRA can estimate the moments

of individual peaks independently ordered by maximum spectral reflectivity. The main difference to a real radar data processing scheme is that the noise $N_i$ is known already in advance. Therefore, the user can choose between using the known $N_i$ or applying the method by Hildebrand et al. (1974) for estimating $N_i$. Based on the noise-corrected radar Doppler spectrum $\eta_v(i)'$, the moments ($Z_e$, $MDV$, Doppler spectrum width $\sigma$, skewness $\gamma$, and kurtosis $\kappa$) and slopes (left, right) of the radar Doppler spectrum are estimated as discussed in (Maahn and Löhnert, 2017). The higher moments and the slopes depend on the

instrument noise, therefore it is crucial to configure PAMTRA in accordance with the radar specifications. All radar moments and the Doppler spectrum are available non-polarized (NN), but if required also for HV (horizontal receive, vertical transmit), VH, VV, and HH polarization. This allows estimation of differential reflectivity ZDR $= Z_e^{HH}/Z_e^{VV}$ and linear depolarization ratio LDR $= Z_e^{HV}/Z_e^{HH}$, among others.

### 2.3 Gaseous absorption

Absorption by atmospheric gases in the microwave range can be separated into contributions by resonant line absorption (i.e., $H_2O$, $O_2$, $N_2$, and $O_3$) and the water vapor continuum. PAMTRA implements various models to calculate the absorption coefficients of atmospheric gases. The model by Rosenkranz (2015) including modifications of the water vapor continuum absorption (Turner et al., 2009) and the line width modification of the 22.235 GHz $H_2O$ line (Liljegren et al., 2005) is selected as default. Alternatively, the Millimeter-wave Propagation Model (MPM93) developed by Liebe et al. (1993) can be used to

simulate the absorption by the gaseous atmosphere. The clear interface structure of PAMTRA gives the possibility to easily include future improvements in gas absorption models, i.e., developments of models in the sub-millimeter wavelength range (Mattioli et al., 2019), as well as the implementation of absorption catalogs (Feist, 2004). This provides the possibility to also account for trace gases, which show abundant but weak absorption features in the microwave frequency range above 200 GHz.

### 2.4 Boundary conditions

The atmosphere is bounded at its upper end by the free space. The radiation emitted by this upper boundary can be described by the cosmic background with its mean radiative temperature of 2.73 K (Fixsen, 2009). The lower boundary of the atmosphere interacting with radiation is the Earth's surface. Thereby, the amount of radiation emitted in each upward direction is defined by the surface temperature and its type, which is determined by setting the emissivity and the model of scattering or reflection. This is not only important for up-welling geometries but also for down-welling in case of a strongly scattering atmosphere

(Kneifel et al., 2010). In PAMTRA, scattering or reflection properties of the surface are estimated assuming either a specular, Lambertian, or Fresnel reflection types (Mätzler, 2006, p. 225). Reflection on natural surfaces can be described by Fresnel equations. For idealized simulations, the emissivity can be fixed. Over land surfaces, PAMTRA makes use of the Tool to Estimate Land Surface Emissivity from Microwave to sub-Millimeter waves (TELSEM[2]; Wang et al., 2017; Aires et al., 2011)





which provides emissivities based on geographic location and time information as angular and frequency dependent monthly mean values based on satellite observations.

The reflection of flat ocean surfaces can also be calculated with the Fresnel reflection formulae. The intensity of the reflection is strongly polarization and angle dependent and characterized by the dielectric properties of the ocean surface as a function of the sea surface temperature and salinity. With the Fresnel reflection formulae, the reflection coefficients and the Stokes reflection matrix can be calculated, as well as the angle and polarization dependent emissivity. Since the reflection and emissivity calculated with the Fresnel relations are valid for calm surfaces and deviate significantly for high wind speeds, corrections for wind speed and therefore sea surface roughness and foam coverage have to be applied. PAMTRA utilizes the Tool to Estimate Sea-Surface Emissivity from Microwaves to sub-Millimeter waves (TESSEM[2]; Prigent et al., 2017). It is based on the community model FAST microwave Emissivity Model (FASTEM; Liu et al., 2011) and is designed for frequencies up to 700 GHz.

## 2.5 Hydrometeor description

PAMTRA has been designed to be flexible considering the treatment of hydrometeors enabling the use of a wide variety of input data. Hydrometeor classes can be defined in a flexible way that allows to exactly match the properties of particles measured, e.g., by in situ microphysical probes or to be consistent with assumptions on PSD, density, shape, etc., made in CRMs. In addition to the assumptions on hydrometeors, the calculation of their interaction parameters (mainly absorption, scattering, and back-scattering) and the integration over the PSD of the specific hydrometeor class is a central part of the RT framework (Johnson et al., 2012).

PAMTRA can handle a flexible number of hydrometeor classes. As an example, for the simulations based on the output of a CRM (see example Sect. 3.2), which provides hydrometeor content and total number concentration for cloud liquid, cloud ice, graupel, snow, rain, and hail the number of hydrometeor classes would be six. For each hydrometeor class, the user can specify their microphysical and scattering properties.

The single particle properties are defined in PAMTRA with respect to the particle maximum extend ($D$) and particle properties, such as PSD, mass–size and velocity–size relation can be easily defined by the user with the help of built-in functions. The user can select either size-resolved distributions of particles directly or the functional form of the PSD.

### 2.5.1 Particle size distribution

Most atmospheric models assume a moment-based microphysical scheme for the treatment of cloud processes. In these schemes, the PSD for each hydrometeor category is assumed to follow a predefined functional form and one or multiple moments of the PSD are simulated as prognostic variables. Using PAMTRA, it is straightforward to ingest the moments of the hydrometeor distributions and reconstruct the full PSD from them.

The PSD forms which are built-in in PAMTRA include the mono-disperse, the inverse exponential, the modified gamma, and log-normal distributions. Some variations of these four main distributions have been implemented to facilitate the interface with some specific weather models. As an example, the formulation used by the two-category ice scheme in the COSMO model





(see Doms et al., 2005, p. 69) assumes mono-disperse distribution where the number concentration is dependent on the ambient temperature. Also the relations reported by Field et al. (2005) and Ryan (2002) relating the inverse exponential distribution parameters to the atmospheric temperature and hydrometeor content are already implemented.

In the PSD construction, one or two PSD parameters are free parameters depending on the settings. PAMTRA derives the
values of the unknown parameters by resolving the system of equations for the moments $M_k$ that are given by the model output. At the current development stage, PAMTRA can use three different quantities related to the PSD moments as input, namely: the total number concentration $N_T = M_0$, the effective radius $r_e = M_3/2M_2$, and the mass mixing ratio $q = aM_b$, where $a$ and $b$ are the parameters of the power-law defining the mass–size relation $m(D) = aD^b$.

PAMTRA can also handle size-resolved distributions of particles giving the largest flexibility in the definition of hydrometeor
content and properties. With this tool, it is possible to set the properties (i.e., mass, area, hydrometeor terminal velocity) of particles for each size range, allowing PAMTRA to ingest in situ observations (see example Sect. 3.4) or the output of numerical models employing size-resolved (binned) microphysical schemes. This flexible interface can also be used to connect PAMTRA with atmospheric models that do not require predefined hydrometeor properties such as those involving the Particle Prediction Properties (P3; Morrison and Milbrandt, 2015) microphysical scheme, or even the semi-Lagrangian super-particle models used
for snow (McSnow; Brdar and Seifert, 2018) or drizzle formation (Hoffmann et al., 2017; Maahn et al., 2019).

### 2.5.2  Single scattering and absorption properties

For liquid hydrometeors, such as cloud droplets, drizzle, or rain drops, the single scattering properties are calculated using Mie theory (Mie, 1908). A large number of refractive index models for liquid water have been published over the last decades (Liebe et al., 1991, 1993; Ellison, 2006, 2007; Stogryn, 1995; Rosenkranz, 2015; Turner et al., 2016). Some of them, such as
Liebe et al. (1991, 1993), are well accepted and very commonly used for liquid water in microwave RT. For liquid water at temperatures higher than $0\,°C$ and the lower frequency range (<150 GHz), the refractive indices of various models are relatively similar. However, for super-cooled liquid water (i.e., liquid water at temperatures below freezing) and higher frequencies, the models increasingly deviate from each other because laboratory measurements of the refractive index in this region are lacking (Kneifel et al., 2014; Cadeddu and Turner, 2011). Recent observations of super-cooled clouds at various sites (Kneifel et al.,
2014) triggered the development of new refractive index models which combine the existing laboratory data set with the new cloud observations (Rosenkranz, 2015; Turner et al., 2016). The model of Turner et al. (2016) is used as the default liquid water refractive index model in PAMTRA. Other models, such as Liebe et al. (1993), Ray (1972), Stogryn (1995), and Ellison (2006), can be chosen by the user in order to allow comparison studies with other RT models or with previous RT simulations.

Frozen hydrometeors, such as ice crystals, snowflakes, or rimed particles, comprise a large natural variability of habits,
densities, and orientations. This variability also affects their interaction with electromagnetic radiation, which explains the still large uncertainties in simulating their radiative properties. As a result, the number of scattering databases with various amount of complexity is rapidly increasing (Kneifel et al., 2018). In PAMTRA, there are several options regarding the definition of particle properties as well as the selection of scattering models. If the ice refractive index is not implicitly included in the selected scattering database, it is calculated using the model by Mätzler (2006).





One of the most widely used approximations for ice and snow particles are spheres or spheroids (Bennartz and Petty, 2002; Petty, 2001; Honeyager et al., 2016; Hogan et al., 2012; Tyynela et al., 2011; Matrosov, 2015). Frozen hydrometeors are usually not composed of a homogeneous medium but rather a mixture of ice, air, or liquid water. Hence, spheroidal approximations always require the calculation of an effective refractive index. In PAMTRA, the generic mixing rule by Sihvola (1989) is used.

It should be noted that differences between various mixing formulas might be significant and we have adopted the mixing rule that Petty and Huang (2009) found to cause the smallest deviation of the scattering properties of spheres when compared with more realistic snowflake shapes. PAMTRA allows to define either a constant density or a size dependent mass–size relation. The scattering properties are then calculated using the cost effective Mie (Mie, 1908) or the more time consuming T-matrix theory (Mishchenko and Travis, 1994); the latter also requires the definition of orientation and aspect ratio of the particles.

The spheroidal approximations and in particular the effective refractive index calculations become increasingly unrealistic as soon as the wavelength becomes similar to the particle size. However, the size at which more complex particle models should be used also depends on the scattering variable (Schrom and Kumjian, 2018). The Discrete Dipole Approximation (DDA; Purcell and Pennypacker, 1973) is considered as a reference method to compute scattering properties of complex shaped particles. An increasing number of databases with various particles and scattering variables have been developed during recent years (Kneifel

et al., 2018). At the moment, the user can select particles of the DDA databases from Liu (2008) and Hong et al. (2009) which provide a number of single ice crystal types as well as a small number of aggregates. In particular for aggregates, PAMTRA also includes the very recent SSRGA for active and passive simulations (Hogan et al., 2017). The SSRGA is a cost effective method to calculate the full phase function representative for an ensemble of aggregates. Unlike the soft spheroidal approximations, no effective refractive index is needed but the fluctuations of mass, which mainly characterize the non-Rayleigh scattering,

are described with a number of coefficients. Those have been derived from a large ensemble of aggregates as described in detail in Hogan et al. (2017). The method is also applicable with reasonable accuracy to light and moderately rimed aggregates (Leinonen et al., 2018). A limitation of the SSRGA is that polarimetric variables cannot be estimated because the interaction of the scattering elements inside the particle are neglected. To our knowledge, PAMTRA is the first RT model which allows to use SSRGA for passive microwave simulations.

## 25  3  Application examples

When developing PAMTRA specific emphasis has been on its ability to interface a broad spectrum of microwave instruments and observing geometries with common atmospheric models and their different output variables and hydrometeor schemes. Here, we demonstrate the high versatility of PAMTRA with a number of application examples based on data from recent field campaigns and state-of-the-art atmospheric models.

All following simulations assume spheres (Mie) for the hydrometeor categories liquid water, rain, graupel, and hail. The SSRGA with the coefficients as in Mason et al. (2019) are used for cloud ice and snow. Herein, using the SSRGA allows us to ensure maximum consistency regarding particle properties such as mass–size relation assumed in the microphysical schemes. The benefit of the SSRGA is illustrated for satellite measurements within a cases study using the Integrated Forecasting





System (IFS) from the European Centre for Medium-Range Weather Forecasts (ECMWF) model output together with passive microwave satellite observations (Sect. 3.1). How PAMTRA can be used for understanding cloud and precipitation processes as well as their representation in the novel ICOsahedral Non-hydrostatic atmosphere model (ICON; Zängl et al., 2015) is shown for both ground-based (Sect. 3.2) and airborne measurements (Sect. 3.3). While PAMTRA is interfaced with the two-moment

microphysical scheme by Seifert and Beheng (2006) for the ICON application, Sect. 3.4 demonstrates PAMTRA's ability to ingest spectrally resolved information - in this case provided by airborne in situ measurements. The code to reproduce the simulations and all following figures is available in the supplement.

## 3.1 Satellite perspective

Microwave satellite observations from polar orbiters, e.g., Advanced Microwave Sounding Unit-A/B (AMSU-A/B) or Mi-

crowave Humidity Sounder (MHS), have provided fundamental insights into tropical storms due to their unique ability to penetrate even opaque cloud systems (Kidder et al., 2000). Furthermore, their assimilation into NWP strongly contributes to forecast skills (Geer et al., 2017), though the assimilation at higher frequencies becomes difficult due to the complex interaction of microwave radiation with frozen hydrometeors in the forward simulation.

To illustrate the benefit of the SSRGA compared to the conventional Mie approach for frozen hydrometeors, we selected

a scene from ex-tropical storm Karl, which has been investigated during the North Atlantic Waveguide and Downstream impact EXperiment (NAWDEX; Schäfler et al., 2018). The ECMWF IFS cycle 41r2 with a 0.1° grid resolution provides the atmospheric input fields for PAMTRA. The IFS applies a one-moment microphysical scheme having four hydrometeor categories with mono-disperse cloud categories liquid and ice and exponentially distributed rain and snow.

Microwave $T_B$ contain frequency dependent contributions from atmospheric gases and hydrometeors which are difficult to

disentangle as can be seen for AMSU-A/MHS measurements (Fig. 2a–c). Liquid clouds and precipitation usually appear as enhanced $T_B$ over radiatively cold surfaces like the ocean (emissivity of 0.5–0.7). Scattering at frozen hydrometeors, i.e., ice, snow, graupel, or hail, lead to a depression in $T_B$ (observed from space), which becomes stronger with increasing frequency. Due to the also increasing absorption by water vapor with higher frequencies, the surface influence is reduced and the scattering effects are better distinguishable from the surface effects (Skofronick-Jackson and Johnson, 2011). In order to illustrate the

scattering effect, three window channels, i.e., 50.3, 89, and 157 GHz were selected. The observed scenes clearly reveal the cyclonic nature of the storm. The occurrence of snow precipitation in its north-easterly sector can be clearly identified by its scattering effect which leads to stronger decreases in $T_B$ with increasing frequency (Fig. 2a–c).

PAMTRA was run twice, once using Mie theory for the calculation of the single scattering properties of cloud ice and snow particles and once using the SSRGA. To match the output with the satellite observation, simulated $T_B$ were convoluted

according to the satellite geometry. Looking at the differences in $T_B$ between observation and both simulations (j–o), it can be seen that especially for the lower two frequencies the simulations show slightly lower values especially in the southern part of the area. At this frequencies, the signal is mainly driven by emission from the surface, the water vapor, and liquid hydrometeors, and not so much by scattering at frozen hydrometeors (Skofronick-Jackson and Johnson, 2011). Therefore, the differences in the $T_B$ can be most likely attributed to an underestimation of the liquid water contents (Fig. 2s) or to the water vapor field




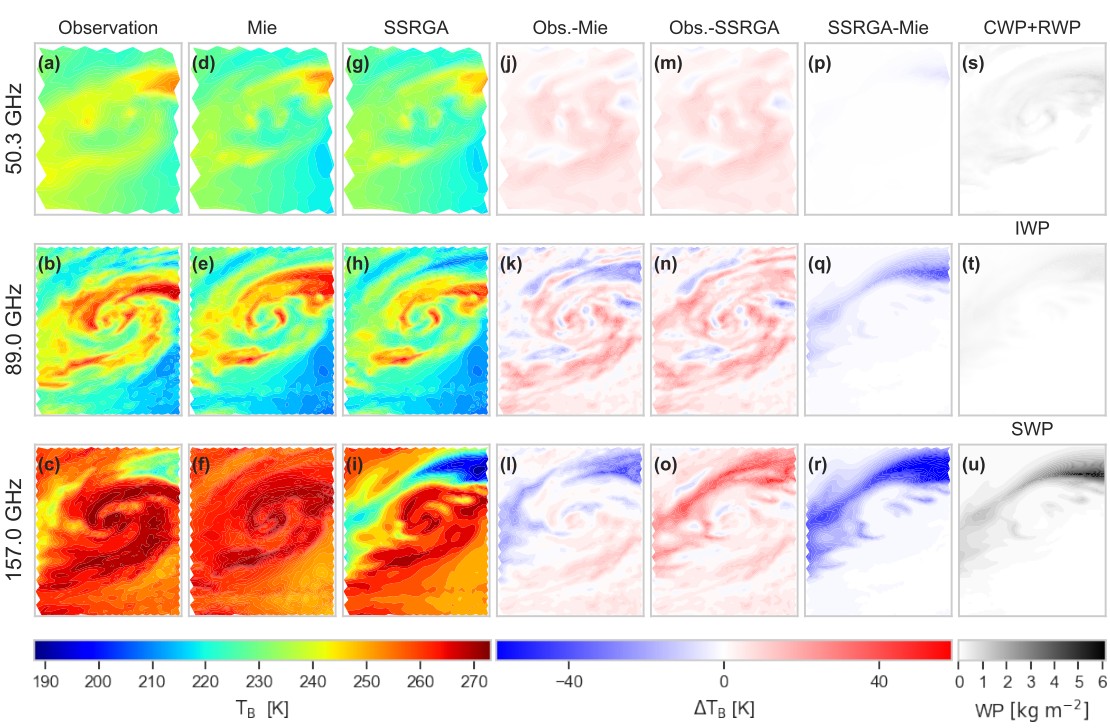

**Figure 2.** Observations with AMSU-A at 50.3 GHz (a) and MHS for at 89.0 (b) and 157.0 GHz (c) for ex-tropical cyclone Karl on 26 September 2016 during the NAWDEX campaign over the North Atlantic; simulations with IFS and PAMTRA with single scattering properties calculated with Mie theory (d–f) and simulations with SSRGA for ice and snow (g–i). Difference in $T_B$ for Obs-Mie (j–l), Obs-SSRGA (m–o), and SSRGA-Mie (p–r). Integrated contents as the sum of cloud water and of rain water path (CWP+RWP) (s), ice water path (IWP) (t), and snow water path (SWP) (u).





(not shown). Differences in the surface signal can be excluded, since the differences are still present at the highest simulated frequency of 157 GHz (Fig. 2l,o), where the surface influence can be neglected. At this higher frequency the scattering at larger frozen hydrometeors becomes more important. In the Mie simulation, a clear underestimation of the scattering effect can be noted as no $T_B$ depression is present in the simulated field (Fig. 2f), although the IFS produces considerable amounts of

snow as shown in the integrated hydrometeor contents with a snow water path up to 6 $kg\,m^{-2}$ (Fig. 2u). The underestimation of the scattering by Mie theory is in accordance with previous studies which find spherical particles introducing a significant positive biases in simulated $T_B$ (Geer and Baordo, 2014). In contrast, simulations with SSRGA (Hogan and Westbrook, 2014; Hogan et al., 2017) are in general able to produce $T_B$ depressions in agreement with the observations. For the simulations shown here based on IFS and PAMTRA using SSRGA for the frozen hydrometeors, the depression is much stronger than for

Mie (Fig. 2r) and comparing it to the observation (Fig. 2o), it can be seen that it is even stronger than in the observation of MHS for the north-eastern area. With the aforementioned capability of SSRGA to reproduce $T_B$ depressions in agreement with observations, this overestimation can be linked to an overestimation of snow water content of ECMWF IFS especially in the middle and upper troposphere.

## 3.2 Ground-based perspective

Novel remote sensing instrumentation combined with high resolution modeling is seen as a way forward to better understand cloud and precipitation processes. In this case study we demonstrate how PAMTRA can be used to simulate a wealth of state-of-the-art ground-based active and passive microwave of observations including radar Doppler spectra at multiple frequencies. The observations shown in Fig. 3 have been recorded on 19 November 2015 as part of the TRIple-frequency and Polarimetric radar Experiment for improving process observation of winter precipitation campaign (TRIPEx; Dias Neto et al., 2019) at the

Jülich Observatory for Cloud Evolution Core Facility (JOYCE-CF; Löhnert et al., 2015). The data have been carefully quality controlled and corrected for radar calibration biases and attenuation by gases and hydrometeors as described in detail in Dias Neto et al. (2019).

The novel ICON model in its Large Eddy version (ICON-LEM; Heinze et al., 2017) with a horizontal resolution of 600 m and 150 vertical layers is used as input to PAMTRA. ICON-LEM is forced by initial and lateral boundary conditions

from the ECMWF IFS. The forward simulations take the different radar specifications (e.g., sensitivity, beam widths, and averaging interval) as described in Dias Neto et al. (2019) into account. $T_B$ are simulated for the 14 channels of a Humidity and Temperature PROfiler (HATPRO; Rose et al., 2005). The passage of a cold front on 19 December 2015 (Fig. 3) is nicely captured by the ICON simulations both regarding vertical and temporal evolution. The first 6 h of model simulations are likely affected by the spin-up of the model (started at 00 UTC) and therefore have been excluded from the figure.

The cloud and precipitation field associated to the cold front causes similar reflectivity structures in the forward simulations as observed. Although the ICON/PAMTRA setup currently does not include a melting layer model, the transition from ice to rain at 1.5–2 km can be clearly seen in the reflectivity and particularly in the mean Doppler velocity (note the well-matched increase in melting layer height). During periods of most intense rainfall (up to 5.6 $mm\,h^{-1}$ between 14 and 15 UTC), attenuation effects are somewhat overestimated in the model but overall the observed signatures are well captured.



Passive observations are unreliable during rainy periods due to potential liquid water on the radome (Cadeddu et al., 2017). During non-precipitating conditions the overall spectral response of the different channels matches the observations very well albeit the high-frequent fluctuations associated with liquid cloud in the observed $T_B$ are missed. This might be due to lower spatio-temporal resolution of the ICON simulation which reproduces the basic temporal evolution but small scale fluctuations

cannot be captured with the resolution used.

A more detailed comparison of modeled and observed microphysical processes is possible due to the ability of PAMTRA to simulate the entire radar Doppler spectrum (Fig. 4). The vertical distribution of the Doppler spectrum during the core precipitation period nicely shows the transition from the slow and narrow ice and snow spectra to wider and faster rainfall at around 2 km height in both simulations and observations. In the ice part, the simulated spectra sometimes reveal bi-modalities

and too large fall velocities (up to $2\,\mathrm{m\,s^{-1}}$). This might indicate some discrepancies in the transition from ice to the snow hydrometeor category in the ICON model or rimed particles in the model with larger fall velocities which are not observed. The observations show at certain heights dynamical effects such as shifting due to vertical air motions or broadening due to turbulence. PAMTRA is principally able to account for these effects if vertical air motion or Eddy dissipation rate is provided. Looking at individual spectra in the ice part (Fig. 4c,d), one can see that the noise levels, shape and velocity region of the ice

Doppler spectra are very well matched. In the rain part (Fig. 4e,f), the Doppler spectra reveal typical resonances at larger drop diameters (first minimum at $6\,\mathrm{m\,s^{-1}}$ corresponding to 1.7 mm size drops (Kollias et al., 2002)) which are also well captured by the PAMTRA forward simulations. The differences in the noise levels (especially Ka-band) are due to known saturation effects in the Ka-band receiver which enhances the spectral noise. The slight mismatch of the W-band noise level is due to height dependent chirp table configuration and associated variable sensitivity (Küchler et al., 2018). In this simulation, PAMTRA was

configured to match the highest chirp sequence and hence the noise level at lower ranges is underestimated.

The ability of PAMTRA to consistently simulate a multitude of radar observables in combination with passive observations provides new opportunities to evaluate microphysics schemes on a process level. For example, the multi-frequency radar observations can be used to distinguish aggregation and riming dominated regions (Kneifel et al., 2015). Additional constraints on e.g. the assumed PSD or terminal velocity–size relation used in a microphysics scheme can be provided by multi-frequency

Doppler spectra (Li and Moisseev, 2019; Kneifel et al., 2016). Finally, the passive observations add information on temperature and humidity profile as well as on vertically integrated liquid water and ice content (Kneifel et al., 2010). PAMTRA is thus not only an important tool to derive new retrievals (Maahn and Löhnert, 2017) but can also be used to develop new microphysical parameterizations as new schemes can be directly confronted with observational characteristics (e.g., typical properties of the radar Doppler spectra).

## 3.3 Airborne remote sensing perspective

Widespread arctic mixed-phase clouds present one of the largest challenges to atmospheric models for weather and climate applications (Morrison and Milbrandt, 2015). Airborne campaigns can provide unique information in this measurement void area. Here we want to demonstrate how airborne active and passive microwave observations can be exploited with the help of PAMTRA simulations to constrain microphysical schemes in CRMs. For this purpose, PAMTRA settings are adapted to

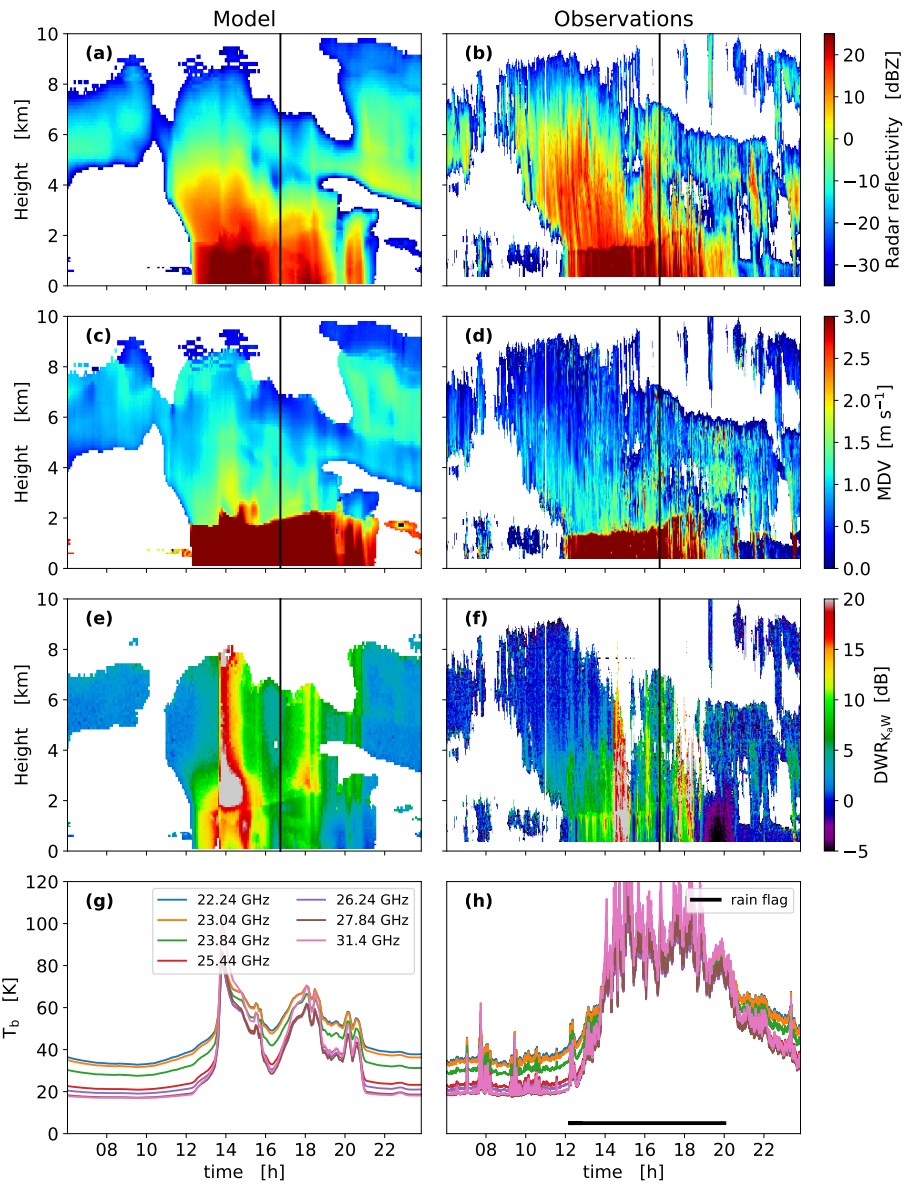

**Figure 3.** Case study of a cold front observed at JOYCE site on 19 November 2015. From top to bottom: time versus height above ground showing equivalent radar reflectivity factor $Z_e$ [dBZ] at Ka-band (a) and (b), mean Doppler velocity MDV [$\mathrm{m\,s^{-1}}$] (c) and (d), dual wavelength ratio DWR between Ka- and W-band [dB] (e) and (f), $T_B$ [K] of the HATPRO microwave radiometer for the seven water vapor (g) and (h). The right side shows the observations, PAMTRA simulations based on the ICON-LEM output are shown in the left column. The black horizontal bar in the HATPRO observation plots indicates periods of active rain flag of the microwave radiometer; data during this period are likely to be disturbed by rain on the radome. The vertical black line in the radar time-height plots indicate the time that is used for the spectra comparison in Fig. 4.

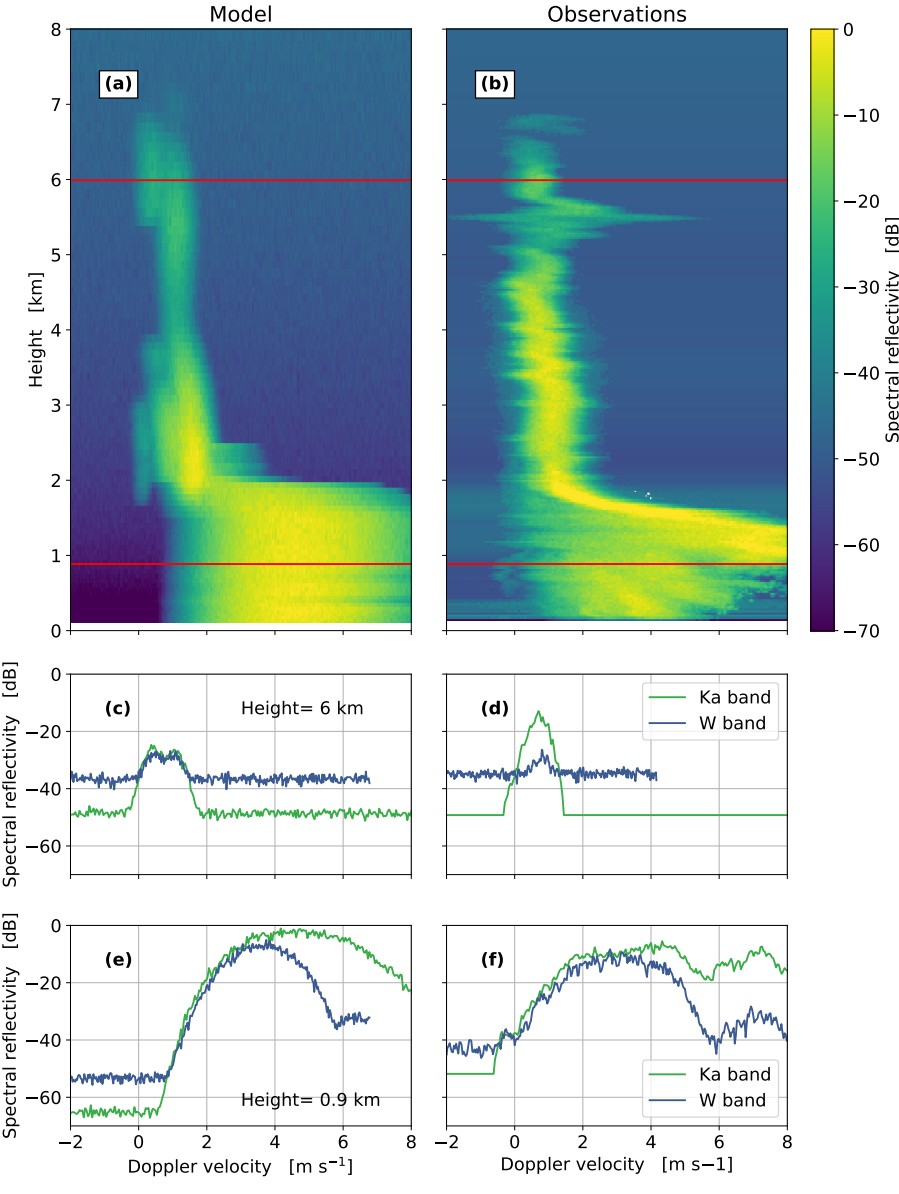

**Figure 4.** Comparison of simulated (left) and observed (right) Doppler spectra at 16:20 UTC of the same frontal case from 19 November 2015 shown in Fig. 3. From top to bottom: Ka-band spectrograms (a) and (b), example Ka- and W-band Doppler spectra in the ice (c) and (d) and the rain (e) and (f) parts of the cloud. The horizontal red lines in the spectrograms indicate the heights at which the example spectra of the successive panels have been extracted.





mimic the measurements of upward directed passive and active radiation made by the airborne Microwave Radar/radiometer for Arctic Clouds (MiRAC; Mech et al., 2019a) flown during the Arctic CLoud Observations Using airborne measurements during polar Day (ACLOUD; Wendisch et al., 2019) campaign. MiRAC combines a 94 GHz frequency modulated continuous wave (FMCW) radar with its integrated passive 89 GHz channel and novel 180–340 GHz radiometer. It was operated aboard

the *Polar 5* research aircraft of the Alfred Wegener Institute, Helmholtz Centre for Polar and Marine Research (AWI) over the Arctic ocean and the sea ice north-west of Svalbard in Mai/June 2017.

The measurements were taken on a flight section of research flight 5 in a cold air outbreak over the Fram street West of Svalbard on 25 May 2017 between 11:30 and 12:00 UTC. The aircraft was flying from West to East over open ocean perpendicular to the atmospheric flow. The reflectivity measurements shown in Fig.5 nicely depict the typical roll cloud structure

that develops when an Arctic air mass transitions from the central arctic to the open ocean during a cold air outbreak (Liu et al., 2006). Their vertical extend is around 750 m and has horizontal length scales of up to [3]km in the observations. By the strong reflectivities in the lowest atmospheric layers, it can be seen that some of the rolls are connected to precipitation, most likely as snow. The enhanced $T_B$ of the 89 GHz passive channel indicate the presence of liquid water over the radiatively cold ocean. A simple regression algorithm for liquid water path (LWP) has been derived from PAMTRA simulated $T_B$ using nearby

dropsondes and artificial clouds, giving an estimate of a maximum LWP of 80 g m$^{-2}$.

Similar to the example shown for ground-based perspectives (Sect. 3.2), the ICON-LEM model was used to simulate the atmospheric conditions this time in a nested approach with the final horizontal resolution of 150 m. Two different simulations have been performed, the first with a standard, fixed vertical profile for cloud condensation nuclei (CCN) and ice nuclei (IN) and second with a parameterization for CCN/IN activation based on (Hande et al., 2016). For the second realization, the fixed

profile was replaced by prognostic CCN and IN and the major part of the change was caused by the activation scheme by (Phillips et al., 2008). The sea surface emissivity is calculated by TESSEM$^2$ based on the ICON-LEM input, i.e., wind speed and sea surface temperature. Gaseous absorption has been calculated according to the Rosenkranz 98 m odel.

As can be seen in Fig. 5, the general structure of the roll clouds with approx. 800 m top height is well captured with the ICON model resolution of 150 m in both simulations. The vertical as well as the horizontal scales of the roll clouds are similar to the

observations. The simulated radar reflectivities with the ICON standard setup (Fig. 5c) are much lower than the observed ones and basically confined to a few 100 m thick cloud layer. Hardly any precipitation reaching the ground is visible in stark contrast to observations. Since the reflectivity at this frequency is mainly driven by large frozen hydrometeors, this indicates too few snow hydrometeors in the simulations. The simulated brightness temperatures agree better with their observational counterpart though a slightly enlarged amplitude - indicating higher LWP - can be seen in the simulations. Looking at the ICON simulation

with modified CCN/IN activation radar reflectivities are generally enhanced compared to the original simulation with maxima of +10 dBZ compared to -20 dBZ in the original run and are now much closer to the observations. As the amplitude of the $T_B$ signal is slightly reduced one can conclude that the modified scheme is able to convert liquid water more efficiently into ice precipitation.

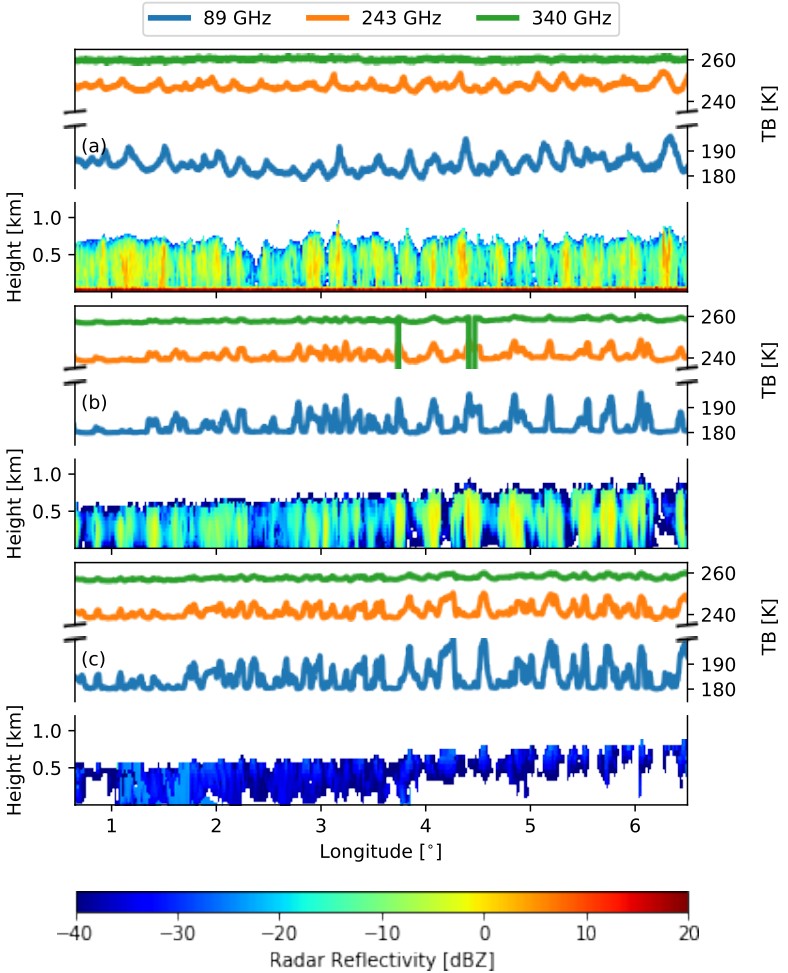

**Figure 5.** Radar reflectivity at 94 GHz and T$_B$ at 89 GHz (blue) with horizontal polarization and 243 (orange) and 340 GHz (green) with mixed polarization as measured by the MiRAC instrument (a) during a 30 min flight section in west-east direction over the Fram street on the 25 May 2017 and simulated radar reflectivity and T$_B$ with ICON-LEM and PAMTRA after improvements made to the microphysical scheme in ICON-LEM (b) and before (c). Note, that radar and 89 GHz measurements were performed under an angle of 25 ° backwards.





For the same scene also coarser resolution ICON standard runs were evaluated which revealed an even larger underestimation of radar reflectivities. While much larger samples of observations and simulations are needed to draw solid conclusions this case study demonstrates the ability of PAMTRA in testing different microphysical schemes and suitability.

### 3.4 Airborne in situ perspective

In the previous examples, typical bulk microphysical schemes were employed in the atmospheric models which implicitly assume a functional relation of hydrometeor properties (e.g., PSD) to which the prognostic model variables, e.g. mixing ratios, number concentrations, can be directly related. New model developments, such as the P3 microphysical scheme (Morrison and Milbrandt, 2015) pose a challenge for RT as their hydrometeor properties, e.g. density, are variable. Similarly, Lagrangian super-particle models (Brdar and Seifert, 2018), models with full-bin microphysics or box models (Hoffmann et al., 2017)

require similar flexibility in the assumptions of hydrometeor properties from the RT. PAMTRA addresses those needs with a full-bin interface (Maahn et al., 2019). In order to demonstrate this feature, we simulate radar Doppler spectra based on airborne in situ observations of liquid clouds. The direct use, i.e., without the need to fit any functional form to the particle properties, of in situ observations in the RT provides numerous possibilities for closure studies between in situ and remotely sensed observations.

The in situ observations of liquid cloud properties have been obtained from the 5th Department of Energy Atmospheric Radiation Measurement (DOE ARM) Program's Airborne Carbon Measurements (ACME-V) campaign obtained at the North Slope of Alaska in Summer 2015. The ARM Gulfstream G-159 (G-1) aircraft of the ARM aerial facility (Schmid et al., 2014, 2016) measured the cloud droplet number concentration for droplets larger than 1.5 μm using a combination of optical cloud probes. The probes and the processing of the cloud probe data set following Wu and McFarquhar (2016) are detailed in Maahn

et al. (2017). In contrast to the other examples, no particle size distribution is assumed but the measured PSD is directly used in PAMTRA through the full-bin interface. Besides the PSD, also the particle mass, density, cross section area, and aspect ratio need to be defined in PAMTRA for every size bin which is trivial for liquid particles. To focus on the idealized development of the spectrum, vertical air motions are not considered in this example.

Figure 6 shows the observed PSD and the resulting radar Doppler spectrum for a vertically sampled cloud at around 22:36

UTC on 27 June 2015. The observed PSD and effective diameter (the ratio of the third and the second moment analogue to the effective radius) show clearly the near-adiabatic increase of droplet size with increasing height caused by condensation. When forward modeled with PAMTRA, this leads to an increase of $Z_e$ because droplet backscattering scales with diameter $D^6$. In the height resolved Doppler spectra (Fig. 6b) this is mainly reflected in the increased spectral reflectivity within the cloud mode with height. Herein, due to the low fall velocity of cloud droplets their Doppler velocities are basically limited to below 0.5

$\mathrm{m\,s^{-1}}$ even close to cloud top where droplets are largest. However, the radar Doppler spectra reveal Doppler velocities up to $1\,\mathrm{m\,s^{-1}}$ in certain heights sometimes showing a clear bi-modality of the spectra, e.g. 800 and 1000 m. These can clearly be attributed to drizzle droplets that are not visible in the in situ measurements (Fig. 6a) as they are rather rare. Their number concentration is likely even underestimated in the ACME-V data set due to the small sampling volume of the used optical probes and thus their influence on radar spectra is likely even underestimated.

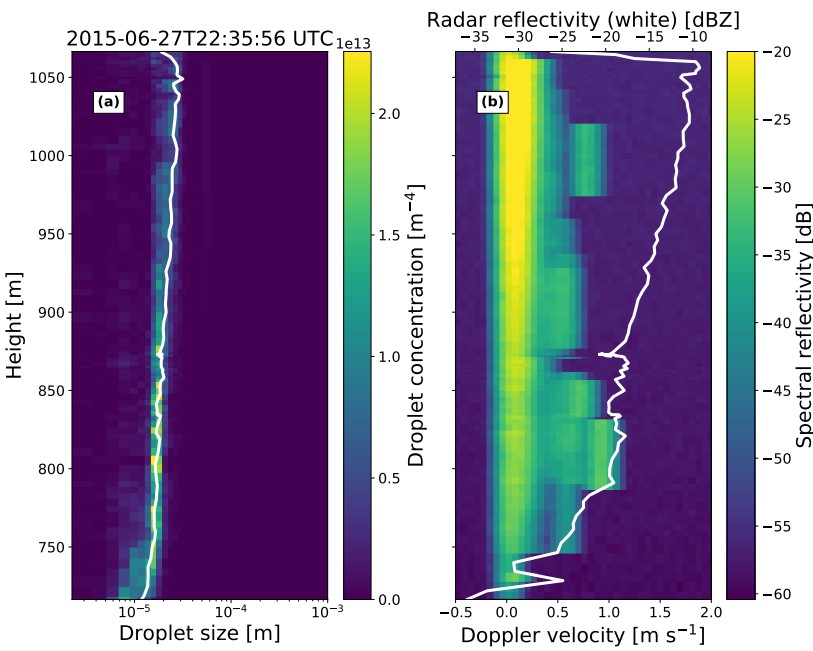

**Figure 6.** (a) Observed drop size distribution and (b) forward modelled radar Doppler spectrum of the sample cloud of the ACME-V campaign. The white line denote (a) the effective diameter and (b) the radar reflectivity $Z_e$.

This example has shown how spectrally resolved information can be exploited by PAMTRA to investigate the impact of different hydrometeors on radar Doppler spectra. Even the simple example of liquid only hydrometeors has shown the high impact of few larger particles on the shape of Doppler spectra. Therefore higher moments of the Doppler spectra, e.g. skewness, kurtosis, contain information which can be used in retrieval algorithms to disentangle the cloud and the drizzle contribution to

5   $Z_e$ (Küchler et al., 2018). Fingerprinting of characteristic hydrometeor signatures in the spectra becomes even more important for frozen hydrometeors allowing detailed process studies (Kalesse et al., 2016).

## 4   Summary and future perspectives

This study introduced the first publicly available version of the PAMTRA forward operator, whose development was motivated by the growing interest to better exploit the unique characteristics of microwave observations in providing information on

10   clouds and precipitation. Specifically, the combination of passive and active microwave sensors on different platforms is very attractive due to their complementary information. To fully exploit the information of the measurements for process studies, the evaluation and further development of cloud resolving models, PAMTRA has been designed as a versatile tool to be compliant with a wide variety of model output and in situ observations as input. Furthermore, PAMTRA aims to support the





ongoing development and application of ground-based instrumentation in particular for multi-frequency radar Doppler spectra measurements, airborne active/passive instrument packages and satellite measurements which will be further extended into the sub-millimeter range, i.e by the Ice Cloud Imager (ICI; Kangas et al., 2014) including channels up to 664 GHz.

PAMTRA simulates the one-dimensional radiative transfer in a plane-parallel atmosphere for polarized passive as well as
the full radar Doppler spectrum for active applications in up- and downlooking mode. PAMTRA has many features already included, i.e., different gas absorption modules, models for the calculation of the surface emissivity, and different methods to calculate the single scattering properties of hydrometeors. Herein, it is unique as the SSRGA can be applied for both passive and active applications. Due to its modularity it can be easily extended when new developments, e.g., new absorption models, single scattering databases, become available. As some applications require massive calcualtions, e.g., databases for retrieval
development or model evaluation, the implementation of parallelization features into PAMTRA supports high performance computing.

Within an example section several applications of PAMTRA as a forward simulator were introduced, which can be reproduced by the interested reader with the help of jupyter notebooks (https://github.com/igmk/pamtra/). The examples consider different geometries, i.e. ground-based, aircraft and satellite, as well as different input sources such as airborne in situ hy-
drometeor spectra, two-moment cloud resolving model simulations and NWP (ECMWF-IFS) analysis. It should be noted that the modular setup of PAMTRA also allows for simpler information such as idealized atmospheric profiles or radiosonde measurements. The latter is especially common for classical retrieval or information content studies for passive microwave measurements such as in (Ebell et al., 2013).

The representation of cloud and precipitation processes is a long-standing problem for atmospheric models and the develop-
ment of new parameterizations and schemes is ongoing in particular for frozen hydrometeors. Microwave scattering by frozen particles provides on the one hand insights into the dominating hydrometeors on the other hand it is also rather challenging due to the wide variety of particles and thus single scattering properties. Progress can only be achieved by the interplay of cloud and RT modeling and its confrontation with measurements. In this context the first example (Sect. 3.1) demonstrates that especially for higher frequencies, i.e. millimeter and sub-millimeter range, the conventional Mie approach is not useful.
PAMTRA can also use the T-matrix approach for the single scattering calculations which, however, requires knowledge on particle orientation. Therefore, the SSRGA approach already used successfully in the radar community has been implemented also for passive RT and shows promising capabilities.

That the forward approach is helpful in disentangling the different contributions of hydrometeors in the measurements is illustrated by further examples. Herein it is important that PAMTRA can be run in high consistency to the models microphys-
ical assumptions. From the ground where novel technologies can be deployed fast multiple frequency radar provides exciting insights into precipitation formation via the ice phase (Sect. 3.2). Airborne measurements allow to reach remote areas such as the Arctic where complex mixed-phase clouds are observed in cold air outbreaks. How active/passive microwave measurements of such clouds can constrain microphysical schemes in the novel ICON model could be shown in Sect. 3.3 by the different response of radar reflectivities and brightness temperatures in respect to the relative contributions by frozen and liquid
hydrometeors.



We did not show an example for the classical observation-to-model approach where data bases of synthetic measurements and corresponding variables of interest are generated for subsequent retrieval development, e.g., (Chaboureau et al., 2008). However, by illustrating how cloud droplet spectra measured by in situ measurements can be used as PAMTRA input for simulating radar Doppler spectra (Sect. 3.4 we could illustrate that higher moments of the spectra can be suitable as retrieval input as

they show clear drizzle signatures (Acquistapace et al., 2019; Küchler et al., 2018). Along this line also the passive microwave signatures of drizzle can be simulated to support related retrieval development (Cadeddu et al., 2019). In general, PAMTRA is well suited for synergetic retrieval development as multitude of microwave measurement quantities, i.e. multi-frequency, polarized brightness temperatures and Doppler spectra moments, can be simulated consistently for the same atmospheric scene.

For the future development of PAMTRA, namely PAMTRA2.0, it is planed to move on to an even more modular code based

on python3 to allow an enhanced parallelization. Further features to be taken into account in the development of PAMTRA2.0 are improvements in the simulation of spectral radar polarimetry, parameterization of frozen surface emissivity as well as simpler adaptations to slant observation geometry. Interested scientists are cordially invited to contribute to the PAMTRA through our online repository.

*Code availability.* The current version of PAMTRA can be found in a publicly available GitHub repository distributed under an GPLv3.0

license found at https://github.com/igmk/pamtra. The exact version of PAMTRA as used for this manuscript is archived on Zenodo (Mech et al., 2019b) including the scripts and data to produce the plots shown in the application section. The code documentation and user manual is compiled into a Read the Docs web page available at https://pamtra.readthedocs.io, and jupyter notebooks that introduce the PAMTRA usage by presenting documented examples and links to the required data are included in the public GitHub repository.

*Author contributions.* M. Mech originally created the PAMTRA model framework consisting of the passive part and the basic methods for

gaseous absorption and single scattering calculations. M. Maahn developed the active radar simulator and the pyPamtra framework and designed the documentation and example framework. M. Mech and M. Maahn are the main authors of PAMTRA. P. Kollias supported the implementation of the radar Doppler spectra simulator in PAMTRA. D. Ori implemented the SSRGA and is strongly involved in the complete development of the single scattering section and the examples. S. Kneifel contributed significantly to the methods of the model for the single scattering calculations, the dielectric properties, and the gaseous absorption. E. Orlandi designed the hydrometeor interface and the particle

size distribution methods. V. Schemann performed the cloud resolving model simulations with the ICON-LEM and is strongly involved in the interfacing of atmospheric models to PAMTRA. S. Crewell contributed to the interpretation of the simulation results and the basic model and manuscript design. M. Mech prepared the manuscript with contributions from all co-authors.

*Competing interests.* The authors declare that they have no conflict of interest.



*Acknowledgements.* The authors like to thank K. Franklin Evans and Graeme L. Stephens for making their model RT4 publicly available.

The authors like to thank as well Heini Wernli for preparing the ECMWF data and Richard Forbes for his support in working with the ECMWF data.

We gratefully acknowledge the funding by the Deutsche Forschungsgemeinschaft (DFG, German Research Foundation) – Projektnummer
268020496 – TRR 172, within the Transregional Collaborative Research Center "ArctiC Amplification: Climate Relevant Atmospheric and SurfaCe Processes, and Feedback Mechanisms (AC)[3]". Partial support for this research was provided by the DFG priority program "High Altitude and Long Range Research Aircraft (HALO)" SPP 1294 "Using the HALO Microwave Package (HAMP) for cloud and precipitation research, grant CR 111/9-1".

This work was also partially funded by the Federal Ministry of Education and Research (BMBF) within the program High Definition
Clouds and Precipitation for advancing Climate Prediction (HD(CP)[2]) under grant 01LK1211.

Contributions by S. Kneifel and D. Ori were funded by the German Research Foundation (Deutsche Forschungsgemeinschaft, DFG) under grant KN 1112/2-1 as part of the Emmy-Noether Group OPTIMIce.

M. Maahn was supported by the US Department of Energy (DOE) Atmospheric Systems Research (ASR) program (DE-SC0013306) and the National Oceanic and Atmospheric Administration (NOAA) Physical Sciences Division (PSD). ACME-V data were obtained from the
Atmospheric Radiation Measurement (ARM) Program sponsored by the U.S. Department of Energy, Office of Science, Office of Biological and Environmental Research, Climate and Environmental Sciences Division. The authors would like to thank Greg McFarquhar and Wei Wu for supporting us with the use of ACME-V data.





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





**Table 1.** Main characteristics and features of PAMTRA.

| | |
|---|---|
| General | Python with Fortran core |
| Setup | 1D, plane-parallel, horizontally homogeneous |
| Geometry | Ground-based, airborne, and spaceborne; vertical and slanted view |
| Frequency range | 1–800 GHz |
| Importers for various sources | GCMs, CRMs, soundings, full-bin models, in situ measurements |
| Surface emissivities | FASTEM, TESSEM$^2$, TELSEM$^2$ |
| Gas absorption | Rosenkranz (1998) (with improvements), Liebe et al. (1993) |
| Dielectric properties of ice | Mätzler (2006) |
| Dielectric properties of liquid | Turner et al. (2016), Ellison (2006), Liebe et al. (1993), Stogryn (1995) |
| Single scattering models | Mie, T-matrix, self-similar Rayleigh–Gans, Liu (2008), Hong (2007) |
| Passive output | Polarized brightness temperatures and radiances |
| Active output | Radar reflectivity and higher moments, polarized Doppler spectrum, LDR, ZDR |