# Peer review of "PAMTRA 1.0: A Passive and Active Microwave radiative TRAnsfer tool for simulating radiometer and radar measurements of the cloudy atmosphere"

_Geoscientific Model Development, 2019_

## Referee Comment (RC1) · Anonymous Referee #1 · 22 May 2020

The manuscript "PAMTRA 1.0: A Passive and Active Microwave radiative TRAnsfer tool for simulating radiometer and radar measurements of the cloudy atmosphere" by Mech et al., presents a new model to simulate polarized microwave measurements with passive or active sensors. The model is highly versatile. It can be applied to different observation geometries and able to be extendeded to new scattering or absoprtion inputs. The original feature of the PAMTRA is its ability compute the Doppler velocity spectrum of radar measurements. Examples of simulations are provided that give a good idea of the model capabilities.

The paper is clearly written and should be published in the journal. I would only like to

raise the following minor points.

1) I think more information are needed about the limitations of the model. I have noticed for examples the following points. What are the limitations due to the "column independent approximation" (P5 L12). What is the maximum range for the elevation angle? Would it be possible to quantify "strong scattering" (P5, L14) and "strong precipitation . . . large radar footprint" (P6, L26).

2) Sect. 2.1: What are the atmospheric input ? (temperature, pressure, humidity, trace gas profiles ?). I think that the radiative transfer equation solved by RT4 should be written. It is the core of PAMTRA for passive observations and it will help to better understand the model simplifications.

3) Sect. 2.2: The pulse width is not discussed for radar simulations. Is-it a model parameter? I think it will have an effect on the spectral width and on the measurement vertical resolution ? Is the latter computed? (I did not see any description of it in the manuscript)

Technical corrections:

P7L21: ".. dynamical and instrument effects such as attenuation . . .". Atmospheric attenuation is not "dynamics" nor "instrument". The sentence should be rephrased.

P8, L15: To my knowledge, N2 does not have resonant lines in the microwave domain but it contributes to the continuum absorption. This contribution is included in a dry continuum term in Liebe together with a contribution from O2. This should be corrected.

P3, Fig.2: What are the spatial coverage and resolution of the maps?

P18,L10: correct [3] in "up to [3]km"

P18L22: correct "model" in "Rosenkranz 98 m odel"

Fig.6 caption: correct "denotes" in "The white line denote . . ."

---

## Referee Comment (RC2) · Anonymous Referee #2 · 27 May 2020

This paper documents a new simulator for passive and active measurements, with a series of test cases illustrating its features. It is a useful documentation of the model and also a nice demonstration of how microwave observations can inform microphysical model developments. The work is in good shape and I only have minor comments.

Bigger minor comments

1) Throughout the examples in section 3, it would be good to have clearer documentation of the atmospheric model (ICON-LEM) and the exact settings of the radiative transfer model (as PAMTRA has a number of options, shown in Table 1).

a) It would be useful to have a short section to centralise the description of, and give

further details on, ICON-LEM. It is important to know the type of microphysics schemes being employed, and which prognostic and active variables are used (e.g. which hydrometeors are represented, and which moments?) Is there any possibility of a mismatch in assumptions (e.g. PSD, shape, fallspeed) between those in PAMTRA and in the model?

b) The PAMTRA settings used in the examples in section 3 need to be more clearly stated. One option might be to extend Table 1.

2) Section 3.1 uses the IFS as input to PAMTRA and compares to AMSU-A and MHS. There are a few issues here:

a) The inputs to PAMTRA likely only include the four prognostic hydrometeors from the large-scale cloud parametrisation (P12 L17). This is insufficient to replicate observed brightness temperatures. In the all-sky forward modelling of passive microwave data at ECMWF, the convective hydrometeors from the convection schem are also included (see e.g. Geer and Baordo, 2014, section 2.2). However these fields are not available from the standard archived ECMWF products. If the convective hydrometeors were added, brightness temperature depressions in frontal areas (which often contain embedded convection) would likely be deeper.

b) This text is overly strong:  "is even stronger than in the observation of MHS for the north-eastern area. With the aforementioned capability of SSRGA to reproduce TB depressions in agreement with observations, this overestimation can be linked to an overestimation of snow water content of ECMWF IFS". The implication from more extensive comparisons in Geer and Baordo (2014) would be that snow water content in the IFS in frontal areas is consistent with observations, at least within the uncertainty on the assumed PSDs and particle shapes in the radiative transfer. The authors seem to be claiming that SSRGA is perfect and the IFS is wrong. It is unlikely that simple, especially given point (a) which, if addressed, would likely make the overestimation of simulated brightness temperature depressions look

even worse when using SSRGA.

Other minor comments

1) The introduction motivates the idea of using remote sensing measurements "for improving the atmospheric models" (e.g. P2 L29). However (P2 L30) when describing the importance of these measurements in data assimilation and NWP, it would be possible to infer that model validation was still their main purpose. It would be worth making it more explicit that the main aim of using these observations in NWP is to infer initial conditions for weather forecasts (the aim to improve models is not yet so well developed in NWP.)

2) P3 L17 suggests that hydrometeor single scattering properties for fast R/T models are derived from line-by-line models, which is not correct. It would be best just to remove the mention of single scattering properties here.

3) P3 L21 I suggest to delete "principally" as it is not clear what this means in the context.

4) P4 L12 -> P6 L2 gives a discussion on horizontal homogeneity, suggesting it is not important in microwave radiative transfer. This is not correct, because the beamfilling effect (due to the nonlinear dependence of backscatter or brightness temperature on water content) means there is ambiguity between water mass and water inhomogeneity at scales below the model grid or sensor field of view. The importance of horizontal inhomogeneity in forward modelling for NWP is illustrated by, among others, Geer et al. (2009) and references therein. However, it's easy to deal with horizontal inhomogeneity by using the independent column approximation. Presumably what the authors really mean is that full 3D radiative transfer with horizontal inhomogeneity is unnecessary.

5) P6 L5 The description of the doubling-adding method in this paragraph is not particularly helpful, and it closely follows the description in Evans and Stephens (1995) which itself doesn't much help summarise the method or ideas like the "interaction principle"

[Figure]

or "initialisation". There might well be a textbook that can help the authors formulate a clearer and simpler description of the technique - is it covered in Petty (2006) or Thomas and Stamnes (2002), for example?

6) P6 L25 The word "However" suggests a dependence between the first part of the paragraph (on the dielectric factor) and the second part (on multiple scattering). In practice these are two completely separate issues, Maybe the second part of the paragraph would be better introduced with "Another issue" rather than "However"?

7) P6 L30 "the minimal sensitivity" - this is unclear and would still be unclear if what the authors mean is "the minimum sensitivity". Is it rather the radar noise that is being referred to?

8) P7 L16 "v_nyq" - is it worth explaining why this parameter is called "nyq" or giving it a simpler notation? (since Nyquist is not mentioned in the text here)

9) Section 2.4 describes the Stokes reflection matrix but is insufficiently clear on how this is being set up, particularly for the components that describe non-specular reflection. For example TELSEM, TESSEM and FASTEM are all emissivity schemes that assume specular and non-polarised reflection at the surface, and provide a simple emissivity to describe this. Yet the text implies they provide a full reflection matrix. There is also an ambiguity as to whether TESSEM is providing just the roughness and foam coverage corrections, or the entire emissivity and reflectivity calculation (P9 L8). A much clearer description is needed here, given that determining the full Stokes reflection matrix (including polarisation changes and non-specular reflections) is not at all straightforward.

10) P9 L23 "particle maximum extend" should have a clearer definition (and "extent", not "extend", is probably intended)

11) P10 L7 consider defining $M_k$ with an equation so that it's easier to understand why $q=aM_b$.

12) P11 L9-10 suggests that the reason to choose Mie or T-Matrix is simply speed - surely it's whether you have a sphere or a spheroid?

13) P14 L2 - the surface is very often visible in satellite 157 GHz observations, outside the humid conditions of the tropics, so errors in the surface representation could very well be suspected here. If the authors want to claim that "the surface influence can be neglected" this would need to be backed up by a map of the surface-to-space transmittances at 157 GHz for this case study.

14) P15 L32-33 - the Arctic is very far from a "measurement void", since polar orbiting operational meteorological satellites cover it with very high temporal frequency. The authors should be more specific on this point.

15) P20 L26 Using "adiabatic" to describe the droplet size variation with height is loose terminology and should be improved - "adiabatic" of course refers to thermodynamic processes, and the radius of water droplets won't change much under a true adiabatic assumption (water being incompressible).

16) P20 L32 and surrounding discussion is initially confusing. It could be more clearly stated in the text that the Doppler spectrum is only simulated, not observed. The suggestion that the larger droplets (secondary peaks in the Doppler spectrum) are "invisible" in the in-situ measurements is confusing as they must be present in the data, just not visible on the colour scale chosen for this plot, or possibly hidden under the white line.

Typos

P4 L14 "plan parallel" -> "plane parallel"

P18 L11 and L22 "extend" -> "extent"

P18 L24 "resulting" -> "resulting simulated"

Bibliography

Only citations not already listed in the bibliography of the paper under review are given here.

Geer, A.J., Bauer, P. and O'Dell, C.W., 2009. A revised cloud overlap scheme for fast microwave radiative transfer in rain and cloud. Journal of applied meteorology and climatology, 48(11), pp.2257-2270.

Petty, Grant William. A first course in atmospheric radiation. Sundog Pub, 2006.

Thomas, Gary E., and Knut Stamnes. Radiative transfer in the atmosphere and ocean. Cambridge University Press, 2002.

———————————————

---

## Author Comment (AC1) · 26 Jun 2020

[12pt,a4paper]article

units

**Anonymous reviewer 2**

We thank the reviewer very much for her/his detailed thoughts, the very useful comments, and suggestions on the manuscript, and thereby the possibility to further improve it. In the following we will first address all "bigger minor comments" and list the changes we made in the manuscript. The minor review points will be answered afterwards. In general, the manuscript has been revised and thereby strengthened according to the reviewers comments. For the more extensive comments in 1a) we have added an appendix to the manuscript.

Text that has been revised or that has been added to the manuscript is written in italic letters.

Bigger minor comments

**1) Throughout the examples in section 3, it would be good to have clearer documentation of the atmospheric model (ICON-LEM) and the exact settings of the radiative transfer model (as PAMTRA has a number of options, shown in Table 1).**

**a) It would be useful to have a short section to centralise the description of, and give further details on, ICON-LEM. It is important to know the type of microphysics schemes being employed, and which prognostic and active variables are used (e.g. which hydrometeors are represented, and which moments?) Is there any possibility of a mismatch in assumptions (e.g. PSD, shape, fallspeed) between those in PAMTRA and in the model?**

Indeed, a description of the Seifert and Beheng microphysical scheme was missing has been added to the manuscript in the subsection of the ground base example (Sec. 3.2) where it used the first time for simulations based on ICON-LEM runs.

*The ICON-LEM model used here implements the 2-moments microphysical scheme from Seifert and Beheng (2006). The cloud scheme has six hydrometeor classes (cloud drops, rain, cloud ice, snow, graupel, and hail) which are assumed to be distributed according to a modified gamma function (Petty and Huang, 2011). The model simulates the evolution of two moments of the hydrometeor distributions, namely the mass mixing ratio q and the total number concentration N. Details on the treatment of the ICON microphysical scheme are given in appendix A2.*

Matching the hydrometeor assumptions in the atmospheric model and PAMTRA is of greatest importance for the accurate simulation of remote sensing measurements. PAMTRA has been designed to be highly consistent with the model assumptions, but is still transparent and easy to use. However, as the user might employ different atmospheric model runs with different assumptions in the end the user needs to check that. For two prominent models (ICON-LEM and IFS) this procedure is described in detail. We have included a new appendix section where all these processing steps are described in greater detail for both the classical ICON and the IFS cycle 41r2 model used in the application examples.

Note that in the manuscript the appendix A1 is referenced for the IFS at the end of the 2nd paragraph of subsection 3.1 as

*...(Forbes et al., 2011) as prognostic variables. More details on the treatment of the IFS microphysics in PAMTRA are given in appendix A1.*

**b) The PAMTRA settings used in the examples in section 3 need to be more clearly stated. One option might be to extend Table 1.**

The information on settings can be found at different parts of the manuscript, very detailed in the accompanying code for the examples, and the general model documentation. In the subsections of section 2, where we describe the options available in PAMTRA for gaseous absorption, models for refractive indices, or surface treatment, we always give the default option if appropriate. None of these default options has been changed in the examples. To make this a bit more prominent, we used bold letters in table 1 for default options and changed the caption of the table to:

*Main characteristics and features of PAMTRA. Default options are written in bold letters.*

Settings important for the presented simulations are mentioned in the manuscript where these are described. More technical settings necessary to run the model for the specific example are shown and described in detail in the jupyter notebooks of each example available with the github project of PAMTRA. Therefore, we kindly like to point to these for answering the comment regarding the settings apart from the ones mentioned in the revised version of the manuscript.

**2) Section 3.1 uses the IFS as input to PAMTRA and compares to AMSU-A and MHS. There are a few issues here:**

**a) The inputs to PAMTRA likely only include the four prognostic hydrometeors from the large-scale cloud parametrisation (P12 L17). This is insufficient to replicate observed brightness temperatures. In the all-sky forward modelling of passive microwave data at ECMWF, the convective hydrometeors from the convection scheme are also included (see e.g. Geer and Baordo, 2014, section**

2.2). **However these fields are not available from the standard archived ECMWF products. If the convective hydrometeors were added, brightness temperature depressions in frontal areas (which often contain embedded convection) would likely be deeper.**

The reviewer is absolutely right. The ECMWF IFS data used in the simulations only include the mass mixing ratios of the four hydrometeor categories as prognostic variables. Additional hydrometeor contents from convective rain and snow flux (personal communication with Richard Forbes) are not available. These, most likely, will influence the resulting brightness temperatures. To account for this insufficiency of the simulations to reflect the observations, this is now mentioned in the manuscript *...rain and snow (Forbes et al., 2011) as prognostic variables. Because the convective rain and snow flux profiles are not available in the standard output, we - in contrast to Geer and Baordo (2014) - can not consider their contribution which may modify the results.*.

**b) This text is overly strong: "is even stronger than in the observation of MHS for the north-eastern area. With the aforementioned capability of SSRGA to reproduce TB depressions in agreement with observations, this overestimation can be linked to an overestimation of snow water content of ECMWF IFS". The implication from more extensive comparisons in Geer and Baordo (2014) would be that snow water content in the IFS in frontal areas is consistent with observations, at least within the uncertainty on the assumed PSDs and particle shapes in the radiative transfer. The authors seem to be claiming that SSRGA is perfect and the IFS is wrong. It is unlikely that simple, especially given point (a) which, if addressed, would likely make the overestimation of simulated brightness temperature depressions look even worse when using SSRGA.**

We agree with the reviewer that our statement was too strong. It is important to note the

it is not our intention to provide an in-depth evaluation of the ECMWF IFS model. This is not possible with the data available and the methods we apply in the comparison. Our intention is more to present an application example of PAMTRA and the tools delivered with it. It is up to the user to make use the toolbox in an appropriate way for tasks like atmospheric model evaluation. To reflect this fact, we weakened our points and made them less conclusive, more speculative, and vague.

P14 L8 up to the end of subsection 3.1 now reads: *For the simulations shown here based on IFS and PAMTRA using SSRGA for the frozen hydrometeors, the depression is much stronger than for Mie (Fig. 2r) and comparing it to the observation (Fig. 2o), it can be seen that it is even stronger than in the observation of MHS for the north-eastern area, although the contributions to the total precipitating hydrometeor amounts through convection are not included in the simulations. With the aforementioned capability of SSRGA to reproduce TB depressions in agreement with observations, this overestimation might be either connected to an overestimation of snow water content of ECMWF IFS especially in the middle and upper troposphere or to an overestimation of the scattering by the SSRGA.*

Other minor comments

**1) The introduction motivates the idea of using remote sensing measurements "for improving the atmospheric models" (e.g. P2 L29). However (P2 L30) when describing the importance of these measurements in data assimilation and NWP, it would be possible to infer that model validation was still their main purpose. It would be worth making it more explicit that the main aim of using these observations in NWP is to infer initial conditions for weather forecasts (the aim to improve models is not yet so well developed in NWP.)**

To our knowledge there are quite some NWP evaluation studies that use are RT models. Therefore we think this is important to include. To make it more clear and to distinguish between these two application examples, we included "or".

*On the other hand, the remote sensing measurements shall be used for improving the atmospheric models, or, most directly, measurements are used in data assimilation together with fast RT operators to infer the initial conditions for NWP models.*

**2) P3 L17 suggests that hydrometeor single scattering properties for fast R/T models are derived from line-by-line models, which is not correct. It would be best just to remove the mention of single scattering properties here.**

This refers to something that has been addressed by the answer to comment 2 of reviewer 1.

**3) P3 L21 I suggest to delete "principally" as it is not clear what this means in the context.**

We deleted "principally".

**4) P5 L12 → P6 L2 gives a discussion on horizontal homogeneity, suggesting it is not important in microwave radiative transfer. This is not correct, because the beamfilling effect (due to the nonlinear dependence of backscatter or brightness temperature on water content) means there is ambiguity between water mass and water inhomogeneity at scales below the model grid or sensor field of view. The importance of horizontal inhomogeneity in forward modelling for NWP is illustrated by, among others, Geer et al. (2009) and references therein. However, it's easy to deal with horizontal inhomogeneity by using the independent column**

**approximation. Presumably what the authors really mean is that full 3D radiative transfer with horizontal inhomogeneity is unnecessary.**

We rephrased this subsection completely. The part where the statement regarding independent column approximation is included new reads the following:

*For the passive part, the one dimensional, polarized, and monochromatic vector RT equation for an azimuthally symmetric scattering media in a plane-parallel atmosphere applying the independent column approximation is solved using the RT4 code of Evans and Stephens (1995). 3D effects can not be modeled but horizontal inhomogeneity can be taken into account by the independent column approximation by realistically describing atmospheric variations along the path (Meunier et al. 2013). The assumption of a plane-parallel geometry is sufficient for most RT problems in the microwave spectral range with the exception of strongly scattering precipitation situations where the radiation does not originate within the instruments field of view (Battaglia and Tanelli, 2011).*

**5) P6 L5 The description of the doubling-adding method in this paragraph is not particularly helpful, and it closely follows the description in Evans and Stephens (1995) which itself doesn't much help summarise the method or ideas like the "interaction principle" or "initialisation". There might well be a textbook that can help the authors formulate a clearer and simpler description of the technique - is it covered in Petty (2006) or Thomas and Stamnes (2002), for example?**

We agree with the reviewers comment, that the description of the doubling and adding method as it is done here is not of much help. Therefore, we decided to reformulate the whole subsection 2.1 to follow as well a suggestion of another reviewer. For the doubling an adding we now point to textbooks that describe the method and do not

give anymore a very rough and non-understandable description. Adding a complete description of the doubling and adding here is beyond the scope of the manuscript.

**6) P6 L25 The word "However" suggests a dependence between the first part of the paragraph (on the dielectric factor) and the second part (on multiple scattering). In practice these are two completely separate issues, Maybe the second part of the paragraph would be better introduced with "Another issue" rather than "However"?**

The reviewer is right, this is not very well written. We changed P6 L25 and the following till the end of the paragraph. It now reads:

*Currently, the simulation of multiple-scattering effects are not implemented in PAMTRA. Multiple-scattering generally increases with the intensity of precipitation, with larger measurement volume, and with increasing radar frequency (Battaglia et al., 2010). For satellite radars, such as CloudSat, multiple scattering effects have to be accounted for in case of heavy precipitation events (Matrosov and Battaglia, 2009). Due to the smaller measurement volume of common ground-based cloud radars, multiple scattering can be usually be neglected.*

**7) P6 L32 "the minimal sensitivity" - this is unclear and would still be unclear if what the authors mean is "the minimum sensitivity". Is it rather the radar noise that is being referred to?**

After re-reading that part we agree with the reviewer that is actually the noise that is important here. We have rephrased the sentence specifying that what affects the radar measurements is

[Figure]

*...the intensity and variance of the spectral noise.*

**8) P7 L16 "vnyq" - is it worth explaining why this parameter is called "nyq" or giving it a simpler notation? (since Nyquist is not mentioned in the text here)**

We changed the beginning of the sentence mentioning the Nyquist frequency to

*The maximum/minimum Doppler velocity of a real radar Doppler spectra is determined by the Nyquist velocity ....*

**9) Section 2.4 describes the Stokes reflection matrix but is insufficiently clear on how this is being set up, particularly for the components that describe non-specular reflection. For example TELSEM, TESSEM and FASTEM are all emissivity schemes that assume specular and non-polarised reflection at the surface, and provide a simple emissivity to describe this. Yet the text implies they provide a full reflection matrix. There is also an ambiguity as to whether TESSEM is providing just the roughness and foam coverage corrections, or the entire emissivity and reflectivity calculation (P9 L8). A much clearer description is needed here, given that determining the full Stokes reflection matrix (including polarisation changes and non-specular reflections) is not at all straightforward.**

TESSEM2 provides the emissivity in the frequency range of 10 to 700 GHz for horizontal and vertical polarization and arbitrary angle from nadir. The parameter range of TESSEM2 is valid for is 10 m windspeed between 0 and 25 m/s, sea surface temperature of 27 to 310K, and salinity between 0 and 40 $°/°°$. As stated by the reviewer, P9 L8 is a misleading and gives the impression, the TESSEM$^2$ only provides a correction. Therefore, we re-phrased the corresponding part, which should as well address the first comment related to emissivity.

*...have to be applied. PAMTRA utilizes the Tool to Estimate Sea-Surface Emissivity from Microwaves to sub-Millimeter waves (TESSEM[2]; Prigent et al., 2017) for the calculation of polarized and foam and roughness corrected emissivities. TESSEM[2] is based on the community model FAST microwave Emissivity Model (FASTEM; Liu et al., 2011) and is designed for frequencies up to 700 GHz. The resulting emissivities and thereby reflectivities, are used to calculate the elements $R_{ij}$ with $i, j = 1, 2$ of the 4-by-4 reflection and radiance matrix needed for solving the radiative transfer by RT4. All other values are set to 0.*

**10) P9 L23 "particle maximum extend" should have a clearer definition (and "extent", not "extend", is probably intended)**

We have specified that the maximum extent is the particle 3D maximum dimension in P9 L23.

**11) P10 L7 consider defining Mk with an equation so that it's easier to understand why q=aMb.**

We have included the definition of a distribution moment as suggested.

**12) P11 L9-10 suggests that the reason to choose Mie or T-Matrix is simply speed - surely it's whether you have a sphere or a spheroid?**

We agree with the reviewer and changed these lines to make it more specific and clear.

*Dependent on the requirements on computational speed and particle properties, the scattering properties can be calculated using Mie (Mie, 1908) or T-matrix theory (Mishchenko and Travis, 1994); for the latter, also the orientation and aspect ratio of*

*the particles have to be defined.*

**13) P14 L2 - the surface is very often visible in satellite 157 GHz observations, outside the humid conditions of the tropics, so errors in the surface representation could very well be suspected here. If the authors want to claim that "the surface influence can be neglected" this would need to be backed up by a map of the surface-to-space transmittances at 157 GHz for this case study.**

The reviewer is right. The argumentation for the influence of the surface signal at 157 GHz is not very good. However, the signal in the microwave region from ocean surfaces is quite well understood. In addition, the IFS SST should not differ that much from observations. Therefore, differences between model and observation that can be attributed to the surface should be rather small.

P13 L33 now reads: *Since the ocean surface signal in the microwave region can be model quite well by TESSEM$^2$ and the sea surface temperature in the model and reality should not differ to much, the differences in the TB can be most likely attributed to an underestimation of the liquid water contents (Fig. 2s) or to the water vapor field (not shown). At the higher frequency of 157 GHz (Fig. 2l,o),, the scattering at larger frozen hydrometeors ...*

**14) P15 L32-33 - the Arctic is very far from a "measurement void", since polar orbiting operational meteorological satellites cover it with very high temporal frequency. The authors should be more specific on this point.**

We changed the text to:

*Airborne campaigns can provide unique information in this area where ground based*

*observations are made at very few stations and where polar orbiting satellites have rather coarse spatial resolution.*

**15) P20 L26 Using "adiabatic" to describe the droplet size variation with height is loose terminology and should be improved - "adiabatic" of course refers to thermodynamic processes, and the radius of water droplets won't change much under a true adiabatic assumption (water being incompressible).**

We removed the "near-adiabatic".

**16) P20 L32 and surrounding discussion is initially confusing. It could be more clearly stated in the text that the Doppler spectrum is only simulated, not observed. The suggestion that the larger droplets (secondary peaks in the Doppler spectrum) are "invisible" in the in-situ measurements is confusing as they must be present in the data, just not visible on the colour scale chosen for this plot, or possibly hidden under the white line.**

We changed that part and hope it is more clear now. It reads:

*However, the radar Doppler spectra reveal Doppler velocities up to 1m s−1 in certain heights sometimes showing a clear bi-modality of the spectra, e.g. 800 and 1000m. These can clearly be attributed to the high impact of low-concentration drizzle droplets on radar observations. These drizzle droplets are not visible in Fig. 6a despite the logarithmic color scale as they are rare. The small sampling volume of the optical probes used during ACME-V leads to poor statistics for drizzle drops which can explain the presence of inhomogeneities in the spectra forward modeled with PAMTRA (Fig. 6.b).*

Typos

**P4 L14** "**plan parallel**" → "**plane parallel**" Changed

**P18 L11 and L22** "**extend**" → "**extent**" Changed

**P18 L24** "**resulting**" → "**resulting simulated**" Changed (but it is P20 L24)

---

## Author Comment (AC2) · 26 Jun 2020

[12pt,a4paper]article

units

[Figure]

**Anonymous reviewer 1**

We thank the reviewer very much for her/his very useful comments and suggestions on the manuscript, and thereby the possibility to further improve it. In the following we will address the more detailed minor comments and list the changes we made in the manuscript. The technical corrections will be addressed afterwards. Text that has been revised or that has been added to the manuscript is written in italic letters.

Minor comments

**1) I think more information are needed about the limitations of the model. I have noticed for examples the following points. What are the limitations due to the "column independent approximation" (P5 L12). What is the maximum range for the elevation angle? Would it be possible to quantify "strong scattering" (P5, L14) and "strong precipitation . . . large radar footprint" (P6, L26).**

To address these points, subsection 2.1 has been revised. It now reads: *For the passive part, the one dimensional, polarized, and monochromatic vector RT equation for an azimuthally symmetric scattering media in a plane-parallel atmosphere applying the independent column approximation is solved using the RT4 code of Evans and Stephens (1995). 3D effects can not be modeled but horizontal inhomogeneity can be taken into account by the independent column approximation by realistically describing atmospheric variations along the path (Meunier et al. 2013). The assumption of a plane-parallel geometry is sufficient for most RT problems in the microwave spectral range with the exception of strongly scattering precipitation situations where the radiation does not originate within the instruments field of view (Battaglia and Tanelli,*

*2011).*

The simulation of the passive radiative transfer at high frequencies for very strong scattering might require that the number of angles to describe the scattering matrix has to be increased. This number is fixed to 16 at the moment in PAMTRA, which is sufficient for most of the applications the model has been applied for so far. For future versions we will give the user the opportunity to adapt this variable, as it is already implemented in the solver backend RT4.

In respect to multiple scattering, we have to stress out again, that PAMTRA is not able to simulate multiple scattering. Whether multiple scattering occurs and whether it needs to be considered for specific situation depends on many different parameters like: considered frequency, beam width, observing geometry, particles, and particle size distribution present, etc.. This is described in more detail by the studies referred to in the manuscript (i.e., Matrosov and Battaglia (2009); Battaglia and Tanelli (2011)).

The radar simulator section has therefore been extended.

*Currently, the simulation of multiple-scattering effects is not implemented in PAMTRA. Multiple-scattering generally increases with the amount of scatterers, with larger measurement volume, and with increasing radar frequency (Battaglia et al., 2010). For satellite radars, such as CloudSat, multiple scattering effects have to be accounted for in case of heavy precipitation events (Matrosov and Battaglia, 2009). Due to the smaller measurement volume of common ground-based cloud radars, multiple scattering can be usually be neglected for this application.*

**2) Sect. 2.1: What are the atmospheric input ? (temperature, pressure, humidity, trace gas profiles ?). I think that the radiative transfer equation solved by RT4**

**should be written. It is the core of PAMTRA for passive observations and it will help to better understand the model simplifications.**

The module to solve the radiative transfer RT4 requires as input profiles of temperature and gaseous absorption at the specified frequency, and if present, profiles of the scattering properties of the hydrometeors for the same frequency. These are calculated (apart from the temperature) by appropriate methods as described in the manuscript. In addition, the type of the surface reflection and emissivity of the surface is needed by RT4. As minimum atmospheric input, PAMTRA needs profiles of temperature and pressure on a height grid. All other values can be either zero or are automatically set to reasonable default values. For these cases a warning is raised.

In our opinion the inclusion of the RT equation does not help the reader or potential user of the model. Including the equation would result in a lengthy explanation of the single terms and contributions which is beyond the scope of this manuscript. To help the interested reader to gain further understanding into the equations behind, we point to the formulation in a more detailed publication by Evans and Stephens (1993, Eq. 2.22) and in Evans and Stephens (1995, Eq.1). Furthermore, we extended the text by describing in more detailed what the assumptions and simplifications of the model are. The whole subsection 2.1 has been completely reformulated-

To make these points more clear in the manuscript, the subsection on the passive radiative transfer has been adapted in addition to what has been mentioned in the answer to comment 1.

*The RT equation is described by the formulation in Eq. 2.22 by Evans and Stephens (1993) or Eq. 1 in Evans and Stephens (1995). It is solved numerically by the doubling and adding method which is formulated and described in detail by several textbooks*

[Figure]

*(i.e., Liou, 2002, p. 290). RT4 requires as input the vertical profiles of temperature and gaseous absorption coefficients and a lower and upper boundary condition. If hydrometeors are present, the profiles of the single scattering properties are required as well. Since a plane-parallel geometry with isotropic thermal emission is considered and all the particles are assumed to be azimuthally random oriented and mirror-symmetric, the radiation fluxes are also isotropic in azimuth. This symmetry in azimuth implies that the third and fourth Stokes components are zero and the RT problem simplifies to the first two components. RT4 does not make use of the Rayleigh-Jeans approximation which relates the Planck function linearly to the brightness temperature is widely used on the microwave regions.*

**3) Sect. 2.2: The pulse width is not discussed for radar simulations. Is-it a model parameter? I think it will have an effect on the spectral width and on the measurement vertical resolution? Is the latter computed? (I did not see any description of it in the manuscript)**

PAMTRA only provides a relatively simple 1D radar simulator, so no beam geometry etc. is considered. Of course, the pulse width affects the vertical resolution. In the model this resolution is defined by the user and his choice of the vertical grid in the atmospheric input. This treatment is reasonable because pulse width and vertical resolution are not strictly tied when pulse compression is used.

In the manuscript we have added at the end of the first paragraph in radar simulator description: *The vertical resolution of the simulated full radar Doppler spectrum is determined by the vertical resolution of the input profiles.*

Technical corrections:

**P7L21: ".. dynamical and instrument effects such as attenuation . . .". Atmospheric attenuation is not "dynamics" nor "instrument". The sentence should be rephrased.**

The sentence has been re-phrased to:

*In reality, the idealized $\eta_v(v)$ spectrum is affected by attenuation, kinematic broadening, vertical air motion, and radar noise (Doviak and Zrnic, 1993).*

**P8, L15: To my knowledge, N2 does not have resonant lines in the microwave domain but it contributes to the continuum absorption. This contribution is included in a dry continuum term in Liebe together with a contribution from O2. This should be corrected.**

The reviewer is right. This was wrong in the manuscript. We re-phrased the sentence to state it in a correct manner.

*Absorption by atmospheric gases in the microwave range can be separated into contributions by resonant line absorption (i.e., H2O, O2, and O3) and the water vapor and dry continuum.*

**P13, Fig.2: What are the spatial coverage and resolution of the maps?**

We changed the figures so that meridians and parallels are included. Given that the ECMWF IFS cycle 41r2 has a resolution of $0.1°$ this shows that the resolution is 6 to 7

km. The spatial coverage is approx. 950 km in North-South direction and 800 and 950 km in Northern and Southern part, respectively.

We added to the manuscript: *...cycle 41r2 with a 0.1° grid (6 to 7 km) ...*

**P18,L10: correct [3] in "up to [3]km"**

Corrected

**P18L22: correct "model" in "Rosenkranz 98 m odel"**

Corrected

**Fig.6 caption: correct "denotes" in "The white line denote . . ."**

Corrected

---

## Author Comment (AC3) · 26 Jun 2020

We had to replace figure 5. Somehow corrupted data have been used for the 340 GHz simulation in the figure 5b). This has been redone. The data change only in that sense that three NaNs were replaced by the crorrectly simulated brightness temperatures. The overall picture and the conclusion didn't change at all.
* * *
[Figure]

**Fig. 1.** Revised figure